# TRANSDUCTION IS ALL YOU NEED FOR STRUCTURED DATA WORKFLOWS

## ABSTRACT

This paper introduces Agentics, a functional agentic AI framework for building LLM-based structured data workflow pipelines. Designed for both research and practical applications, Agentics offers a new data-centric paradigm in which agents are embedded within data types, enabling logical transduction between structured states. This design shifts the focus toward principled data modeling, providing a declarative language where data types are directly exposed to large language models and composed through transductions triggered by type connections. We present a range of structured data workflow tasks and empirical evidence demonstrating the effectiveness of this approach, including data wrangling, text-to-SQL parsing, and domain-specific multiple-choice question answering.

## 1 INTRODUCTION

Large language models (LLMs) have demonstrated remarkable capabilities in natural language understanding, reasoning, and tool use. Recent advances in LLM-based agent systems—equipped with human-level text generation and conversational abilities—have opened promising directions in software engineering, scientific research, and a wide range of tasks that can be automated (Hosseini & Seilani, 2025; Agashe et al., 2025). In the emerging paradigm of agentic AI, LLMs are integrated with external tools, structured knowledge sources, and memory modules to form specialized agents. These agents, each designed with distinct functions and behaviors, collaborate through modular reasoning to solve complex tasks (Acharya et al., 2025; Moshkovich & Zeltyn, 2025; Huang & Huang, 2025; Han et al., 2024).

Despite growing interest in agentic AI, current systems remain poorly suited for structured data workflows, where inputs and outputs are governed by explicit schemas and semantics (Hopkins et al., 2022). Embedding structured data into natural language often fails in enterprise use cases such as analytics and data transformation (Sarirete et al., 2022; Heck, 2024; Putri & Athoillah, 2024), due to the lack of compositional guarantees, which leads to brittle workflows, cascading errors, and limited reproducibility.

While existing frameworks (LangChain, 2025; CrewAI Inc., 2025; Pydantic, 2025b; Dibia et al., 2024; Khattab et al., 2024) offer modular agent composition and tool integration, they fall short of addressing the foundational challenge. Namely, how to endow agentic systems with algebraic structure that ensures robustness, modularity, and interpretability. As a result, these pipelines remain fragile, i.e., lacking formal semantics and struggling with structured data integration.

To overcome these limitations, we introduce `Agentics`, a framework for agentic AI grounded in *logical transduction algebra*, a formalism for representing and composing transformations between structured inputs, intermediate states, and outputs. `Agentics` provides a unifying programming model for generative structured data workflows, treating each step as an asynchronous transduction rather than a prompt-chained interaction. This abstraction enables modularity, parallelism, and schema-constrained transduction, addressing the fragility and lack of formal semantics in existing agentic pipelines.

At the core of `Agentics` lies the notion of logical transduction: a typed transformation that maps an input object of one schema into an output object of another. The key distinction is that `Agentics` treats agents as stateless transducers operating over well-defined data types, hence shifting away from chat-or event-based multi-agent architectures toward a data-centric, functional pipeline. Unlike

```
class ProductReview(BaseModel):      class SentimentSummary(BaseModel):
    reviewer: str                        sentiment: Literal["positive",
    text: str                                             "neutral", "negative"]
    stars: int                           reason: str
```

```
{"reviewer": "Alice", "text": "Excellent    {"sentiment": "positive",
    product quality and fast delivery!", "    "reason": "Excellent quality and fast
    stars": 5},                               delivery"},
{"reviewer": "Bob", "text": "It's okay, but  {"sentiment": "neutral", "reason": "Okay
    the package was damaged", "stars": 3},      product, but package issues"},
{"reviewer": "Carol", "text": "Terrible      {"sentiment": "negative",
    experience, broken after one use!", "    "reason": "Broke after one use"
    stars": 1}                               }
```

Figure 1: Logical Transduction Applied to Sentiment Summary

conversational agents that rely on serialized dependencies and multi-turn dialogue, `Agentics` agents support fully asynchronous execution.

By design, `Agentics` exposes a programmatic interface to LLMs in which all input and output data are represented as typed objects, ensuring schema validation and constraint checking. Interestingly, such structured types are particularly well handled by LLMs, and align with function-calling patterns, making them particularly well-suited for reliable and interpretable inference(Rossiello et al., 2021; 2023). Figure 1 illustrates a simple example of logical transduction. A `ProductReview` object containing a reviewer's name, review text, and rating is transduced into a new object that includes a sentiment label and a rationale. The LLM generates these new fields based on the structured input and the schema of the target type, no additional prompt engineering is required. Beyond individual transductions, `Agentics` introduces an asynchronous *map-reduce style* programming model for asynchronous workflow composition, enabling scalable, and controllable pipelines. In `Agentics` programming model, every step is a typed transformation, enabling reproducibility and adaptability in real-world applications.

Section 2 summarizes the related works and highlight the key difference between `Agentics` and existing frameworks. Section 3 develops logical transduction algebra and asynchronous programming model, Section 4 develops technical implementation of the proposed framework as a Python library, Section 5 shows experiment results on a wide variety of tasks that are closely related to generative structured data workflow tasks such as data wrangling Narayan et al. (2022) and text-to-SQL semantic parsing (Hendrix et al., 1978; Androutsopoulos et al., 1995). We also evaluate schema rich domain-sensitive multiple choice question answering tasks.[1]

In this paper, we focus on proposing a new programming model for agentic AI, rather than introducing novel algorithms or engineered solutions aimed at improving task-specific performance. Nevertheless, our evaluation shows that `Agentics` achieves competitive or improved results due to the structured prompting compared to carefully selected baselines. Our emphasis is on the advantages of shifting toward a data-centric paradigm, demonstrating how this approach simplifies the development of LLM-based workflows while enhancing compositionality, scalability, and execution efficiency.

## 2 RELATED WORK

For decades, AI has been framed as an effort to emulate human intelligence. The Turing Test exemplifies this anthropomorphic ideal: a machine is deemed intelligent if its conversation is indistinguishable from a human's. The rise of LLMs has amplified this framing. By enabling rapid prototyping of intelligent systems through natural language prompts, developing AI agents has become more intuitive and accessible. Consequently, many agentic AI frameworks adopt agentic metaphors such as memory, planning, and tool use, often realized via chat-based interfaces.

This approach has driven remarkable progress in consumer applications. Yet, as tasks demand greater semantic precision, prompt-centric methods often prove brittle, opaque, and hard to scale, especially in

---

[1]Due to the space limitation, we provide details of implementation of each task and additional tasks in the Appendix.

structured data environments where reproducibility and accuracy are paramount. Agentic frameworks typically position the agent as the locus of intelligence, with data as passive input. While effective for open-ended tasks, this model struggles in deterministic workflows. In enterprise settings where querying, transformation, and integration of structured data are common, conversational agents can introduce error propagation and unpredictable behavior.

To address the limitations of prompt-centric agentic systems, recent research has introduced techniques such as guardrails (Ouyang et al., 2022; Dong et al., 2024; Zhang et al., 2024), self-reflection (Shinn et al., 2023; Asai et al., 2024), and correction strategies (Madaan et al., 2023; Pan et al., 2024). While these methods enhance reliability, most frameworks still depend on free-form text or loosely structured prompts that remain fragile and difficult to verify, especially in tasks requiring high semantic precision. In practice, techniques that offer more reliable interfaces such as structured decoding based on predefined schemas (Rossiello et al., 2023; Kwon et al., 2023a) or type-safe libraries (Pydantic, 2025a) have gained traction. These approaches are increasingly supported by modern software stacks built around LLMs.

The shift toward structured data computation is reflected in several emerging frameworks. `Pydantic-AI` (Pydantic, 2025b) emphasizes type-safe agentic programming and serves as agent framework for building structured AI applications using Pydantic types and multi-agent systems. `LangGraph` (LangChain, 2025) enables orchestration of stateful agents over finite state machines, supporting complex control flows. `CrewAI` (CrewAI Inc., 2025) demonstrates strong performance in multi-agent coordination and message passing between agents and tools. `DSPy` (Khattab et al., 2024) pioneers declarative abstractions for prompt engineering and optimization, tightly coupled with structured templates. While these frameworks have evolved toward conversational multi-agent coordination in networked environments, `Agentics` proposes a shift toward computation centered on data semantics and type-driven transformations—enabling more robust, scalable, and interpretable workflows for generative structured data tasks.

## 3 LOGICAL TRANSDUCTION ALGEBRA

`Agentics` leverages asynchronous, parallel LLM inference to support enterprise-scale workflows over structured data. To ground this capability, we introduce *Logical Transduction Algebra* (LTA): a typed, compositional calculus for building, analyzing, and optimizing LLM-powered pipelines. Our work is closely related to relational algebra (Codd, 1970) and the MapReduce programming model (Dean & Ghemawat, 2008). Here, we present an abridged version of the formal Logical Transduction Algebra in the main paper. The full details are provided in the Appendix.

A *logical transduction* is a semantically grounded transformation from an object $x$ of type $X$ to an object $y$ of type $Y$ such that each field of $y$ is logically justified by information in $x$ under the constraints of the source/target schemas. Concretely, schemas are realized as `Pydantic` types in Python library implementation, which allows type-checking and constraints make intermediate states explicit and auditable, while aligning naturally with LLM function-calling behaviors. Logical transduction is executed between any two Agentics (AGs).

**Definition: Agentics (AG)**  Let $\Theta$ be the universe of types. A type $T \in \Theta$ is a finite set of named slots $T = \{(s_i, T_{s_i})\}$ with $T_{s_i} \in \Theta$. An *Agentic structure $AG$* bundles a schema and a list of instances:

$$AG := \{ \, s_{\text{atype}} : \Theta, \quad s_{\text{states}} : \text{List}[ \, s_{\text{atype}} \, ] \, \}.$$

We write $AG[X]$ for an agentic structure with schema $X$ and $\mathbf{x} = AG[X]$ for a particular instance list. Concatenation of state lists endows instances of a fixed $AG[X]$ with a monoid structure, giving a simple but useful algebra over batches.

**Definition: Transduction operator ($\ll$)**  The basic operator of LTA is the left-shift $\ll$, which maps a source object into the target schema:

$$\mathbf{y} := AG[Y] \ll x \quad \text{where} \quad \mathbf{y}.s_{\text{states}} = \{ \, y \, : \, y \text{ satisfies } Y \text{ and is logically inferred from } x \, \}.$$

Operationally, $\ll$ renders typed inputs into prompts, invokes an LLM (optionally with tools/RAG/few-shot), and parses/validates the result into the target type $Y$.

**Definition: Prompt function (P)**   A prompt function $P : \text{List}[T] \rightarrow \texttt{str}$ serializes typed states to text, bridging structured data and LLM inputs. Zero-shot transduction applies $P$ per state:

$$\mathbf{y}[i] \;=\; AG[Y] \ll P(\mathbf{x}[i]),$$

. The default prompt function of AG is the pydantic model_dump() method with returns a json dictionary representing the state.

**Lemma: Properties of LTA**   Let the *transduction context* , i.e. the LLM, decoding settings, tools, and few-shot used by the AG, fixed. Then the following conditions applies:

- *Conditional determinism*: Re-invoking $\ll$ on the same $x$ under the same context yields the same $y$, enabling reproducibility.
- *Statelessness*: $y$ depends only on $x$ and the context, not on other inputs, enabling asynchronous parallel execution.
- *Compositionality*: If $\mathbf{y} = AG[Y] \ll \mathbf{x}$ and $\mathbf{z} = AG[Z] \ll \mathbf{y}$, then $\mathbf{z} = AG[Z] \ll AG[Y] \ll \mathbf{x}$, giving functional-style pipeline composition.

**aType Operators**   In LTA, operations among agentic types (aTypes) provide the foundation for composing and reasoning about structured workflows. Each type X is defined as a set of named slots $(s_i, T_{s_i})$, and standard set operations are applied component-wise: the union $X \cup Y$ collects all slots present in either schema, the intersection $X \cap Y$ keeps only shared slots, the difference $X \setminus Y$ removes slots from X that also appear in Y, and the Cartesian product $X \times Y$ builds composite types pairing slots from both. At the level of instances, Agentic structures AG[X] (a schema plus a list of states) form a monoid under concatenation of state lists, with the empty instance as identity.

**Asynchronous MapReduce.**   To scale beyond single-step transduction, LTA provides two higher-order operators:

$$\texttt{aMap} : (AG[X], f) \rightarrow AG[Y], \qquad \texttt{Reduce} : (AG[X], g) \rightarrow AG[Y],$$

where $f : X \rightarrow \text{List}[Y]$ is applied independently to each $x_i$ (filter/transform/fan-out), and $g : \text{List}[X] \rightarrow Y$ aggregates many states (summaries, rankings, joins). Because $\ll$ is stateless, `aMap` execute the function f, which might embed transductions, in asynchronous fashion. In contrast, `Reduce` functions takes all the states of the input agentic at once, and therefore cannot be asyncronously executed.

In short, LTA provides a formalism to make assumptions explicit; $\ll$ provides a uniform *contract* between stages; `aMap`/`aReduce` expose parallel structure and enable hierarchical summaries; and conditional determinism/statelessness support reproducible, high-throughput execution. Together, LTA turns LLM pipelines from brittle prompt chains into modular, optimized, and auditable programs.

## 4   TECHNICAL IMPLEMENTATION

`Agentics` framework is distributed as a Python library As briefly introduced in the logical transduction algebra, we define the Agentic structure AG as a container for a list of typed objects. Within this framework, LLM inference is conceptualized as logical transduction, i.e., the process of inducing one object from another based on a predefined type schema. The framework is built upon two essential components: the Agentic structure, implemented as the metaclass AG, and the Transducer, built on top of `Pydantic` to enable type-safe object transformation. In this section, we describe these core elements and present examples that illustrate the Agentic programming model in practice.

### 4.1   EXAMPLE USAGE OF META-CLASS AG

The meta-class `AG` is a lightweight yet expressive container for typed data and its execution context. It provides the following three essential capabilities. First, it binds a *typed schema*, which we call `atype`, a `pydantic.BaseModel` subclass. Second, it holds a *list of states*, where each `state` is a validated instance of `atype`. Third, it carries an *execution context of transducer*, such as `llm`, tools, prompt template, batch size, and decoding parameters.

**Meta-Class AG**   The meta-class `AG` allows structured states to be natively represented in Python, while remaining agnostic to the specific LLM providers (e.g., OpenAI, Google DeepMind, Anthropic, Meta AI, WatsonX, etc). The following pseudo code shows an example of `AG`.

```python
class AG(BaseModel):
    atype: Type[BaseModel]                    # target schema (Pydantic model)
    states: List[BaseModel] = []              # instances of 'atype'
    # execution context
    llm: Any = None                           # LLM client/handle
    tools: Optional[List[Tool]] = []          # optional tool registry
    prompt_fn: Optional[Callable] = lambda x: x.model_dump()
    batch_size: int = 20                      # async batch size
```

Given an `atype` and a list of typed instances, an instance of `AG` carries the `states` and manages their transformation either through logical transduction or via asynchronous functions in a map/reduce-style program.

**Pydantic Models as Schemas**   An `atype` can be any subclass of `Pydantic BaseModel`. For instance, a news schema `StockNews` may be defined as follows, and an `AG` class `news` can be instantiated with states of type `StockNews`.

```python
class StockNews(BaseModel):
    ticker: str    # e.g., "AAPL"
    content: str   # raw news article text
    date: datetime # publication time

news = AG(atype=StockNews, states=[StockNews(ticker="AAPL", content="
    Apple shares surged after record iPhone sales were announced.",date="
    2025-01-15")])
```

Note that `AG` instances behave like a typed list enriched with a type system.

```python
news = AG.from_csv("apple_news.csv")
for article in news:     # iterate states like a list
    print(article.ticker)
news.add_attribute("sentiment", str)
news.rebind_atype(StockNews) # rebind to new schema with added attribute.
```

**Logical Transduction (≪)**   `Agentics` framework overloads the ≪ operator to express *logical transduction* between agentic structures. The left operand is the *target* AG, which becomes populated by transducing the states of the right operand. The following examples shows application of logical transduction to sentiment summary by populating the `SentimentScore` instances from `news`.

```python
class SentimentScore(BaseModel):
    ticker: str
    sentiment: float   # normalized score in [-1, 1]
    label: str         # "pos", "neutral", "neg"
output = await (AG(atype=SentimentScore) << news)
#output[0]: SentimentScore(ticker="AAPL", sentiment=0.8, label="pos")
```

**Map/Reduce Paradigm**   The meta-class `AG` applies asynchronous mapping to its list of `states`, executing either logical transduction or a (possibly asynchronous) function $f$ via `AG.amap(f)` on each `state` independently. Since transductions are stateless, `amap` evaluations over multiple states can run asynchronously in parallel. In other words, the mapping can be batched for efficiency. In contrast, `AG.aReduce(f)` aggregates a collection of states into a single (or small set of) output(s). Because it consumes the entire list of states, it operates synchronously and is not parallelizable.

**Example Workflow**   By chaining logical transduction (≪), `aMap`, and `aReduce`, a complex workflow can be expressed declaratively. For example, a sentiment-driven stock ranking can be built

as follows. This combination of typed schemas, logical transduction, and asynchronous map–reduce execution yields workflows that are modular, interpretable, and highly scalable.

```python
# (1) Gather news for each ticker
news_ag = await AG(atype=StockNews).amap(fetch_news_for_ticker)
# (2) Extract sentiment per article
scores_ag = await AG(atype=SentimentScore) << news_ag
# (3) Aggregate to stock-level sentiment
stock_sentiment = scores_ag.aReduce(group_by_ticker_mean)
# (4) Rank portfolio by sentiment
portfolio_ranking = stock_sentiment.aReduce(rank_portfolio)
```

## 4.2 PYDANTICTRANSDUCER

Logical transduction triggers the creation and execution of a `PydanticTransducer`, which is a stateless agent whose role is to generate a valid instance of the target `atype` given text input. The textual input can represent virtually any concept accessible to an LLM, while the structured output ensures reliability in downstream tasks. Because agents in `Agentics` avoids shared conversational memory, it naturally supports asynchronous execution and efficient scale-out.

To ensure high throughput and responsiveness across datasets of varying sizes, the transduction operator processes data in configurable batches, typically ranging from 8 to 32 items. As shown in the figure on the right, execution time decreases as batch size increases from 1, with performance gains saturating around a batch size of 32. This trend reflects near-linear improvements in processing speed across all tasks, highlighting the efficiency of batched execution. The underlying asynchronous map/reduce programming model—grounded in a logical transduction algebra—enables scalable and robust computation, adapting seamlessly to diverse data workloads.

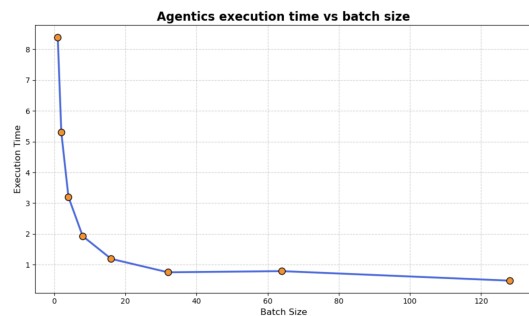

Figure 2: Average time (sec) per question

## 5 EXPERIMENTS

In this paper, we evaluate Agentics, a programming framework for building generative data workflows with a focus on computational efficiency, scalability, ease of design, and accuracy. While prior work on agentic AI frameworks has emphasized capabilities such as planning and tool use, we focus on the quality and structure of agentic workflow pipelines, particularly in data-centric tasks. We test the following hypotheses: (1) Agentics supports a data-centric paradigm through declarative data modeling via type schemas, which decouples logical agent workflows from chat-centric paradigms. This enables intuitive and functional composition of pipelines., (2) Structured prompts induced from declarative data models reduce the burden of manual prompt engineering while maintaining or improving task performance. They are effective in tasks that benefit from clear contexts.

### 5.1 DATA WORKFLOW TASKS

We evaluate canonical structured data workflow tasks such as schema matching with healthcare data (Parciak et al., 2024b), data imputation with the Buy and Restaurant datasets (Narayan et al., 2022), and text-to-SQL with challenging benchmarks (Zheng et al., 2024; Li et al., 2023).

### 5.1.1 SCHEMA MATCHING

Schema matching is a canonical data workflow task that identifies mappings between semantically identical elements in two relational schemas. Recent works such as (Parciak et al., 2024a) have explored the use of LLMs for the schema matching task. The schema matching benchmark by Parciak et al. (2024a) contains 9 datasets, each consisting of a source table from the MIMIC-IV dataset Johnson et al. (2023) and a target table from the OHDSI OMOP Common Data Model OHDSI (2019). Specifically, the mapping problem involves identifying which source column can be matched to a column in the target table. We provide details of the experiment in the Appendix. The schema of the source and target table follows the following data model with table and column names and descriptions. `Attributes` defines a list of `Attribute` states.

```python
class Attribute(BaseModel):
    relation_name: Optional[str] = Field(None, description="table name")
    relation_description: Optional[str] = Field(None, description="table description")
    attribute_name: Optional[str] = Field(None, description="column name")
    attribute_description: Optional[str] = Field(None, description="column description")
class Attributes(BaseModel):
    attributes: Optional[list[Attribute]] = Field(None, description="list of Attributes")
```

**Logical Transduction**    Given a source table with $M$ columns and a target table with $N$ columns, we implemented the mapping task as described in (Parciak et al., 2024a) in two variations: *1-to-1* and *1-to-N*. In the *1-to-1* setting, each prompt contains a pair of columns from the source and target tables, and the LLM is tasked with determining whether they are semantically equivalent. This setting requires $MN$ LLM calls. In the *1-to-N* setting, each prompt contains a column from the source table and $N$ columns from the target table, requiring only $M$ LLM calls. The LLM evaluates all possible pairs and identifies all semantically equivalent matches in a single inference.

The following pseudocode shows the Agentics program that creates Agentics objects for the source and target tables. To parallelize the mapping, `mimic_omop` creates a product of the two Agentics objects and adds an `invertible` attribute to transduce the truth assessment of the mappings.

```python
mimic_data = AG.from_states(mimic_states, Attribute)
omop_data = AG.from_states(omop_states, Attributes)
mimic_omop = mimic_data.product(omop_data)
mimic_omop = mimic_omop.add_attribute(slot_name = "invertible", slot_type = list[bool])
mimic_omop = await mimic_omop.self_transduction(mimic_omop.fields, ["invertible"])
```

**F1-score Result**    We have implemented the schema matching task in Agentics and ensembled results from two open-source models, GPT-OSS and LLaMA-4.[2] The mean F1-score over 9 datasets is summarized as follows. For the *1-to-1* setting, the GPT-3.5 baseline achieves 0.241, while Agentics achieves **0.325**. For the *1-to-N* setting, the GPT-3.5 baseline achieves **0.398**, and Agentics achieves 0.382, slightly behind.[3] Note that we achieved significantly better performance with smaller-parameter models in the *1-to-1* setting, and slightly worse results in the *1-to-N* setting.

### 5.1.2 TEXT-TO-SQL

We evaluated text-to-SQL pipelines composed of multiple components on challenging text-to-SQL benchmarks. The dev set of BIRD-bench (Li et al., 2023) contains 1,534 questions across 11 databases, and the dev set of the Archer dataset (Zheng et al., 2024) contains 104 questions across 10 databases.[4] In our evaluation, we focus on the data-centric design aspects of Agentic text-to-SQL pipelines. Composing various components such as few-shot examples, schema linker outputs, keywords from topic models, and sub-questions with optimized prompts—collectively improves execution match results by 10.33% over the baseline performance of Llama-3.3-70B.

---

[2]Ensembling the two pipelines from different models takes the disjunction of their assessments, which may decrease accuracy but increase recall.

[3]Detailed F1-scores are available in the Appendix.

[4]State-of-the-art performance on both benchmarks is available on the official leaderboards. The best dev performance is 76.14% on BIRD-bench (https://bird-bench.github.io/) and 38.46% on Archer (https://sig4kg. github.io/archer-bench/).

**Asynchronous Pipeline**    The text-to-SQL task can introduce the following data model for each problem in the dataset, where a natural language `question` and the SQL `ddl` scripts for creating the database tables are the input fields provided, and the remaining fields are generated by passing through asynchronous pipelines. The `enrichment` field annotates semantic meanings for the tables and columns, `sql_query` is the generated SQL from LLMs, and `execution_result` is the resulting data frame obtained by executing the `sql_query`.

```
class Text2SQLTask(BaseModel):
    question: str = Field(description="The input natural language question")
    ddl: str = Field(description="The database schema in DDL")
    enrichment: Optional[DB] = Field(description="Additional database enrichments")
    few_shots: Optional[Problem] = Field(description="question, sql pair")
    sql_query: Optional[str] = Field(description="SQL query generated")
    execution_result: Optional[List[Dict[str, str]]] = Field(description="resulting table")
```

The asynchronous pipeline concatenates the logical transductions in order, as shown in the pseudo-code below. A series of logical transductions generates the empty fields in the `Text2SQLTask` state object. Note that the overall execution of all state objects is performed concurrently, either per transduction step or per line of the pseudo-code.

```
text2sql("enrichment.keywords")<<text2sql("enrichment.description")<<text2sql("ddl_schema")
text2sql("enrichment.subquestions") << text2sql("question")
text2sql("enrichment.linked_schema") << text2sql("enrichment.ddl_schema", "enrichment.
    subquestions", "question")
text2sql("few_shots.question", "few_shots.sql_query")<<text2sql("ddl_schema", "enrichment.
    description")
text2sql("few_shots") = text2sql.filter(valid_sql, "few_shots")
text2sql("sql_query") << text2sql("question", "enrichment", "few_shots")
text2sql("execution_result") = text2sql.amap(execute_sql_query)
```

**Brid-Dev Result**    We evaluated execution accuracy on BIRD-dev using a simple prompt and a composite workflow that includes additional logical transductions with randomly generated few-shot examples (**FS**), keyword enrichment (**KW**), sub-question enrichment (**SQ**), schema linking enrichment (**SL**), and an optimized prompt template (**OP**). Across all models, the composite workflow consistently improved performance. Llama-3.3-70B showed the largest gain, improving from $50.51\% \pm 0.71$ to $60.84\% \pm 0.53$, a **10.33%** increase. Mistral-Large improved by **8.21%**, and Llama-4-maverick-17B saw a more modest gain of **3.06%**. These results demonstrate that structured prompting and modular transductions in Agentics can significantly enhance execution accuracy. Inspecting the impact of each individual component, as shown in the Appendix, **SL** and **OP** contributed 2.24% and 4.49% individually, while other components did not improve accuracy on their own. However, combining all components results in a higher gain of 10.33%, exceeding the sum of individual improvements.

**Archer-Dev Result**    The Archer benchmark presents challenges that require LLMs to perform commonsense, arithmetic, and hypothetical reasoning in order to correctly generate SQL expressions from natural language questions. In our experiments, we adopt a simplified pipeline that bypasses intermediate components and directly generates the `sql_query` from the `ddl` and optional `commonsense_knowledge` hints.

We evaluate three LLMs, GPT-OSS-120B, Llama-3.3-70B, and Llama-4-17B under two conditions, with and without commonsense knowledge. Based on the average execution match over 10 trials, GPT-OSS-120B shows a clear improvement when commonsense knowledge is incorporated, increasing from 0.28% to 0.35%. Llama-3.3-70B performs identically in both settings 0.15%, and Llama-4-17B also improved execution match performance from 0.25% to 0.30%. For reference, the best reported performance on the Archer dev set is 38.46% using the GPT-o1 model. Our results with GPT-OSS-120B demonstrate comparable performance, highlighting its effectiveness despite the simplified pipeline and smaller parameter models.

### 5.1.3  DATA IMPUTATION

The data imputation task is also a canonical example in generative structured data workflows. We consider the task of filling in missing entries with plausible substitutions for a given tabular record containing one or more missing field values. In `Agentics`, the input record can be defined as a semantic data type. The missing attribute is then transduced from the known attributes using logical

transduction. We evaluate data imputation implemented in `Agentics` on the Buy and Restaurant datasets Narayan et al. (2022). The zero-shot accuracy performance of the GPT-OSS-120B and Llama-4-17B models is 70.77% and 72.31% on the Buy dataset, and 79.07% and 66.28% on the Restaurant dataset, respectively. These results are comparable to the baseline performance of GPT-3-175B, which achieves 84.5% on Buy and 70.9% on Restaurant.

## 5.2 DOMAIN-SPECIFIC MULTI-CHOICE QUESTION ANSWER

We evaluate the *FailureSensorIQ* dataset, a recently proposed domain-specific multiple-choice QA benchmark designed to assess understanding of sensor relationships and failure modes (Constantinides et al., 2025). We demonstrate the stable performance of structured prompting in `Agentics` across both the original single-correct MCQA and its perturbation variants. The single-correct MCQA set contains 2,667 questions spanning various industrial assets. The perturbations are designed to be knowledge-invariant. The *simple perturbation* involves renaming option letters, while the *complex perturbation* combines option letter renaming with question rephrasing.

**Logical Self Transduction**   Each problem in *FailureSensorIQ* dataset has multiple fields, answer `options`, industrial `asset_name`, `relevancy` context, `question_type`, `subject` of the question, and the `answer` to the question. The structured prompting compose the input prompt from the data model and the Agentics framework performs self-transduction from all other fields to the `answer` with additional instruction as follows.

```
1  fsiq_benchmark = await fsiq_benchmark.self_transduction(
2      input_fields=["question", "options", "option_ids",  "asset_name", "relevancy", "
           question_type", "subject"],
3      output_fields=["answer"],
4      instructions=("Read the input questions, all possible answers, and background task
           information. This is a multiple choice test, where one of the options is true and the
            others are false. Select the answer with the highest likelihood of being correct,
           and return it along with a confidence score and a verbal assessment explaining your
           judgment."))
```

**Accuracy and Robustness Result**   We evaluated accuracy on 2,667 *FailureSensorIQ* instances across four models with varying parameter sizes. Agentics consistently improved performance over the baseline prompt performance for all models. Qwen3-8B showed the largest gain, improving from 45.86% to 60.18%. Llama-3.3-70B and Mistral-Large saw gains of **9.04%** and **8.32%**, respectively. The largest model, Llama-3-405B, showed a modest improvement of **1.64%**, Notably, `Qwen3-8B` achieves a major improvement of **14.32%**, placing it just behind `openai-o1`, 60.4%. These results suggest that prompting through logical transduction helps unlock latent reasoning capabilities, even in models with limited parameter counts.

`Agentics` demonstrates robustness against knowledge-invariant perturbations. In the original `FailureSensorIQ` experiments, all models experienced significant drops in performance, ranging from 5% to 20% . However, `Agentics` showed minimal change, ranging from just 0.08% to 0.19% under the simple perturbation, and from 2.21% to 2.44% under the complex perturbation.

## 6 CONCLUSION

We present a principled framework for agentic AI, grounded in a novel logical transduction algebra and a scalable asynchronous programming model. This framework redefines how agents interact with data through a declarative, type-driven approach, enabling robust and efficient execution across diverse tasks. Despite its strengths, the current framework has several limitations. First, it primarily focuses on type-driven transduction, which may not generalize well to tasks requiring richer contextual understanding or instruction-following behavior. Many real-world tasks involve implicit signals or external context that go beyond type annotations. Second, the integration of tool usage remains underexplored. Future work will explore several promising directions. One is transduction that incorporates instruction or retrieval. Another is enhanced tool integration, enabling agents to invoke external tools within the transduction pipeline. Additionally, extending the framework to support interoperability with other agentic AI frameworks could unlock broader capabilities of agentic systems.

**Ethics Statement** Our approach does not involve human subjects, personally identifiable information, or synthetic data generation that could be misused. We do not deploy models in production settings, and all evaluations are conducted with open source dataset and open weight models. On the usage of large language models. We used large language models to polish the writing for fixing syntax errors or latex command errors.

**Reproducibility Statement** We provide several materials to ensure the reproducibility of our work. First, we have anonymized the Python library code and included it as supplementary material. The library can be installed locally, and the provided examples are sufficient to reproduce the experiments. In the Appendix, we include pseudocode for the experiments described in the paper, which aids in understanding the implementation details. We use publicly available open-source datasets, and all data sources are freely accessible. To support data preprocessing, we also provide the relevant scripts—such as those used for schema matching and data imputation experiments. We also provide details about the computing resources and the large language models used in our experiments, which are documented in the Appendix. For the theoretical contributions related to logical transduction algebra, we include the full version with complete proofs in the Appendix.

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

# A EXPERIMENTS DETAILS

## A.1 COMPUTING INFRASTRUCTURE

In the experiments, we benchmark open-weight instruct tuned models ranging from larger or smaller parameter version of GPT-OSS-120B, Llama-3.3, Llama-4, Qwen-3, and Mistral. For the experiment that measures the running time, we host LLMs in local vLLM (Kwon et al., 2023b) server with four A-100-80GB GPUs for Llama-3-3-70B model, and one A-100-80GB GPU for other 8B parameter models.

- In the Text-to-SQL experiment, we used GPT-OSS-120B, Llama-3.3-70B, Mistral-Large, and Llama-4-Maverick-17B models. These models were also run in a cloud computing environment.
- In the schema matching and data imputation experiments, we used GPT-OSS-120B, and Llama-4-Maverick-17B models. These models were also run in a cloud computing environment
- In the Domain Specific MCQA experiments, we used instruction-tuned models such as Qwen3-8B, Llama-3.3-70B, Mistral-Large, and Llama-3-405B. We locally hosted the Qwen3-8B model to measure running time, while the other three models were used in a cloud computing environment.
- In automatic prompt optimization experiment, we locally hosted Qwen3-8B and Llama-3.3-70B models.

The following shows the parameters for hosing Qwen3-8B and Llama-3.3-70B models with vLLM (Kwon et al., 2023b).

```
1  GPUS=4
2  CPUS=16
3  MEM=200GB
4  MODEL="meta-llama/Llama-3.3-70B-Instruct"
5  LEN=16000
6    vllm serve ${MODEL} \
7    --max-model-len ${LEN} \
8    --tensor-parallel-size ${GPUS} \
9    --gpu-memory-utilization 0.9
10
11 GPUS=1
12 CPUS=8
13 MEM=64GB
14 LEN=8000
15 MODEL="Qwen/Qwen3-8B"
16 vllm serve ${MODEL} \
17    --max-model-len ${LEN} \
18    --tensor-parallel-size ${GPUS} \
19    --gpu-memory-utilization 0.9
```

Listing 1: vLLM parameters and computing resources

## A.2 TEXT-TO-SQL

We evaluated various text-to-SQL pipelines on the challenging Bird-bench (Li et al., 2023) -dev dataset. We observe that by composing various components such as few-shot examples, schema linker outputs, keywords from topic models, and sub-questions with optimized prompts, each model significantly improves execution match results, achieving up to a 10.33% increase over the baseline performance of Llama-3.3-70B.

Previously, a summary result table of individual experiments were shown, whereas here we expand on the performance by conducting and aggregating multiple runs (5) against the benchmark. This includes setting a high model temperature of 0.9, thus diversifying the generated transduction samples.

The resultant Table 1 highlights the performance gains on average by including additional transductions on top of the base prompt. We note that every individual transduction (i.e. FS / KW / SQ / SL / OP) does not in fact improve average performance on all models. However, techniques such as schema linking and prompt-optimization yield greater improvements on a few models than the other approaches. Interestingly, when including all the transductions together (i.e. FS + KW + SQ + SL + OP), performance significantly improves in a manner that is greater than the sum of its parts. This result indicates that models can be pushed into greater performance by stipulating the right prompt programs to captures the task. Having a simple modeling framework, like `Agentics`, allows for more sophisticated augmentations and feedback loops to improve the creation of such programs.

Future attempts at modeling more complex prompt programs include the addition of feedback and self-correction loops by running failed samples on the database and re-prompting. Additionally, model-dependent optimizations can be illuminated upon, since the performance of the executed program is dependent on the model itself and should therefore be guided by the model more closely.

| Method | Model | | |
|--------|-------|--------------|---------------------------------------|
| | llama-3-3 70b-instruct | mistral-large | llama-4-maverick 17b-128e-instruct-fp8 |
| P | $50.51 \pm 0.71$ | $47.20 \pm 0.78$ | $53.88 \pm 1.37$ |
| P+FS | $50.32 \pm 0.52$ $(-0.19)$ | $45.63 \pm 0.9$ $(-1.57)$ | $53.47 \pm 0.56$ $(-0.41)$ |
| P+KW | $50.32 \pm 0.32$ $(-0.19)$ | $45.09 \pm 0.81$ $(-2.11)$ | $53.03 \pm 0.82$ $(-0.85)$ |
| P+SQ | $51.58 \pm 0.58$ $(+1.07)$ | $46.15 \pm 0.44$ $(-1.05)$ | $52.33 \pm 0.62$ $(-1.55)$ |
| P+SL | $52.75 \pm 0.07$ $(+2.24)$ | $49.54 \pm 1.23$ $(+2.34)$ | $50.46 \pm 0.05$ $(-3.42)$ |
| P+OP | $54.90 \pm 0.53$ $(+4.39)$ | $48.93 \pm 0.5$ $(+1.73)$ | $53.15 \pm 0.65$ $(-0.74)$ |
| P+FS+KW+SQ+SL+OP | $60.84 \pm 0.7$ $(+10.33)$ | $55.41 \pm 1.2$ $(+8.21)$ | $56.94 \pm 0.23$ $(+3.06)$ |

Table 1: Execution accuracy on BIRD-dev, testing a prompt **P** with the simplified task vs. additional transductions in the composite workflow. **SL** replaces the full DDL schema with a linked schema, **KW** includes keyword topic modeling, **FS** randomly generates sql-validated few-shot question-query pairs, **SQ** extracts sub-questions, and **OP** optimizes the prompt template.

The following Table 2 summarizes the evaluation result on Archer dataset.

| Model | With Commonsense | Without Commonsense |
|-------|------------------|---------------------|
| GPT-OSS-120B | $0.35\% \pm 0.02$ | $0.28\% \pm 0.02$ |
| Llama-3.3-70B | $0.15\% \pm 0.01$ | $0.15\% \pm 0.01$ |
| Llama-4-17B | $0.30\% \pm 0.02$ | $0.25\% \pm 0.032$ |

Table 2: Execution accuracy on Archer-dev. We evaluted 104 dev problems with and without commonsense knowledge provided in the dataset.

### A.3 SCHEMA MATCHING

Schema matching is an important task in data integration which is defined as the task of identify mappings between semanticaly identical elements in two relational schemas such that they refer to the same real world concepts. Recent works such as Parciak et al. (2024a) have explored the use of LLMs for the schema matching task.

In this section, we present an experiment that demonstrates how the schema matching task can be formulated as a transduction using the Agentics framework.

**Dataset**: we have used the benchmark datasets from Parciak et al. (2024a) which contains 9 datasets each containing a source table from MIMIC-IVdataset Johnson et al. (2023) and a target table from

OHDSI OMOP Common Data Model OHDSI (2019). True matches between the columns of the two tables are annotated in the ground truth. Table 3 illustrates the 9 benchmark datasets with their corresponding table names and columns.

Following the same patterns as in Parciak et al. (2024a), we have implemented the task in two variations: *1-to-1* and *1-to-N*. In the *1-to-1* setting, each prompt contains a pair of columns from the source and the target table, and the LLM is tasked to determine if they are semantically equivalent or not. In the *1-to-N* setting, each prompt contains a column from the source table and N columns from the target table. The LLM is tasked with matching all possible pairs and identifying all semantically equivalent pairs in a single inference.

| Dataset | MIMIC | | OMOP | | Candidates | GT Matches |
|---|---|---|---|---|---|---|
| | table name | columns | table name | columns | | |
| AdCO | admissions | 16 | condition_occurrence | 16 | 256 | 2 |
| AdVD | admissions | 16 | visit_detail | 19 | 304 | 5 |
| AdVO | admissions | 16 | visit_occurrence | 17 | 272 | 8 |
| DiCO | diagnoses_icd | 5 | condition_occurrence | 16 | 80 | 2 |
| LaMe | labevents | 10 | measurement | 20 | 200 | 10 |
| PaPe | patients | 6 | person | 18 | 108 | 5 |
| PrDE | prescriptions | 17 | drug_exposure | 23 | 391 | 6 |
| SeVD | services | 5 | visit_detail | 19 | 95 | 5 |
| TrVD | transfers | 7 | visit_detail | 19 | 133 | 6 |
| **Total** | | | | | **1839** | **49** |

Table 3: The list of datasets with the name of the table from MIMIC and OMOP schemas along with the number of columns in each table. GT matches illustrate how many true schema matches are present between the columns of the given tables.

Results: Table 4 illustrates the results of the experiments. We have implemented the schema matching task in Agentics and enesembled results from two open source models gpt-oss and llama4.

| Datasets | 1-to-1 | | 1-to-N | |
|---|---|---|---|---|
| | GPT 3.5 Baseline | Agentics gpt-oss & llama4 | GPT 3.5 Baseline | Agentics gpt-oss & llama4 |
| AdCO | 0.000 | 0.500 | 0.133 | 0.250 |
| AdVD | 0.000 | 0.330 | 0.083 | 0.250 |
| AdVO | 0.235 | 0.400 | 0.320 | 0.143 |
| DiCO | 0.667 | 0.500 | 0.800 | 0.667 |
| LaMe | 0.471 | 0.364 | 0.500 | 0.500 |
| PaPe | 0.571 | 0.000 | 0.500 | 0.667 |
| PrDE | 0.222 | 0.000 | 0.417 | 0.211 |
| SeVD | 0.000 | 0.500 | 0.400 | 0.400 |
| TrVD | 0.000 | 0.333 | 0.429 | 0.600 |
| **Mean** | **0.241** | **0.325** | **0.398** | **0.382** |

Table 4: Schema matching F1 scores. GPT 3.5 baseline results are from Parciak et al. (2024a).

## A.4 DATA IMPUTATION

Data imputation constitutes a critical task in the remediation of incomplete or noisy datasets. For a given record in tabular format containing one or more missing attribute values, the objective is to reconstruct the missing entries with plausible substitutions.

Data imputation serves as an illustrative use case for demonstrating the capability of the proposed framework in handling structured input–output tasks with LLMs. Specifically, the input record -

excluding the missing value- can be modeled as a semantic type, while the output can be represented as an extension that compels the LLM to predict the missing entry by leveraging the attributes of the input type as contextual information.

We evaluate our method on the Buy and Restaurant datasets Narayan et al. (2022). In the Buy dataset, given a product name and its description, the model is tasked with predicting the manufacturer as the missing value. In the Restaurant dataset, the goal is to infer the type of restaurant based on its name, address, and phone number.

The results of our framework in zero- and few-shot settings are presented in Table 5.

| Dataset | Model | zero-shot | few-shot |
|---------|-------|-----------|----------|
| Buy | gpt-oss | 70.77 | 70.77 |
| | llama-3.3 | 49.23 | 58.46 |
| | llama-4 | 72.31 | 75.38 |
| Restaurant | gpt-oss | 79.07 | 80.23 |
| | llama-3.3 | 67.44 | 48.84 |
| | llama-4 | 66.28 | 47.67 |

Table 5: Accuracy results for different models and few-shot settings on Buy and Restaurant data imputation tasks.

### A.5 DOMAIN-SPECIFIC MULTI-CHOICE QUESTION ANSWER

FailureSensorIQ benchmark (Constantinides et al., 2025) is recently proposed domain-specific multiple choice QA benchmark to test LLMs' ability to reason about failure modes and sensor relationships. The leaderboard shows that the best performing `openai/o1` model scores 60.4%.

**Dataset** We evaluated 2,667 single-correct MCQA instances spanning various industrial assets, with questions around identifying the right sensor which can detect a given failure mode for a given asset, or identifying the right failure mode that a given asset and sensor can detect. This requires nuanced understanding of sensor behavior, failure propagation, and asset-specific operational logic, and performing logical deductions across the different knowledge about the asset. An example query may be:

```
1  {
2    "Question": "For electric motor, if a failure event rotor windings fault occurs, which
       sensor out of the choices is the most relevant sensor regarding the occurrence of the
       failure event?",
3    "Options":["A. partial discharge", "B. resistance",  "C. oil debris", "D. current", "E.
       voltage"]
4  }
```

**Methods** Baseline results are obtained from the leaderboard that evaluates the standardized prompts with at most three trials for invalid responses.

Our approach leverages the `Agentics` framework to perform schema-constrained transduction from structured input to structured output. Each input instance is represented using a subset of fields from the FailureSensorIQ schema—specifically, the question, asset name, option ids, options, and subject—which are sufficient to ground the reasoning process in both linguistic and domain-specific context.

To improve inference efficiency, `Agentics` supports *parallel batch execution* via the `aMap` operation. This distributes multiple structured prompts across concurrent model invocations, significantly reducing total runtime. Unlike sequential prompting, which processes one question at a time, batch transduction enables scalable evaluation and deployment.

**Experimental Configuration** We evaluate four models ranging from 8B to 405B parameters: **Qwen3-8B** (Yang et al., 2025), **Llama-3.3-70B-Instruct** (Touvron et al., 2023), **Mistral-Large-Instruct-2407** (Jiang et al., 2024), **Llama-3-405B-Instruct** (Touvron et al., 2023). Models are tested using both the original FailureSensorIQ baseline pipeline and the `Agentics` framework. The

baseline uses loosely formatted natural language prompts and retries up to three times if the output is invalid. In contrast, `Agentics` uses structured prompting and schema-constrained decoding. To measure execution time, we host `Qwen3-8B` on a dedicated node with an `A100 80GB GPU` running `VLLM` (Kwon et al., 2023b). Other models are accessed via cloud computing platform. We vary batch sizes to assess scalability and throughput.

**Accuracy Improvement** Table 6 shows the accuracy comparison. `Agentics` improves the performance on all evaluated models, with smaller models benefiting the most. Notably, `Qwen3-8B` achieves a major improvement of +14.32%, getting right behind `openai-o1`, 60.4%. This suggests that prompting through logical transduction helps unlock latent reasoning capabilities, even in models with limited parameter counts.

| Model | # Params | Baseline | Agentics |
|---|---|---|---|
| Qwen3-8B | 8B | 45.86 | 60.18 (+14.32) |
| Llama-3.3-70B | 70B | 41.69 | 50.73 (+9.04) |
| Mistral-Large | 123B | 50.09 | 58.41 (+8.32) |
| Llama-3-405B | 405B | 51.26 | 52.90 (+1.64) |

Table 6: Accuracy (%) of on 2,667 FailureSensorIQ instances.

**Running Time** Figure 3a illustrates the average time (sec) per question for `Qwen3-8B` across varying batch sizes. As shown, parallel batch execution yields substantial speedups, from 8 seconds per question at batch size 1 to less than 1 second per question at batch sizes greater than 16. This improvement is nearly linear as the batch size increases from 1 to 4, after which it begins to saturate.

**Perturbation Study** We follow `FailureSensorIQ`'s perturbation study using the `Agentics` framework to study if there are any robustness benefits that comes with the structured workflows that our framework offers. We experiment with the following knowledge invariant perturbations:

- Option letter renaming; changing the option letters from of A., B., C., to other letters like P., Q., R. We'll call this "Simple" Perturbation.
- Option letter renaming and question rephrasing done by an LLM. We'll call this "Complex" Perturbation.

We use the already prepared perturbed datasets from the original paper.

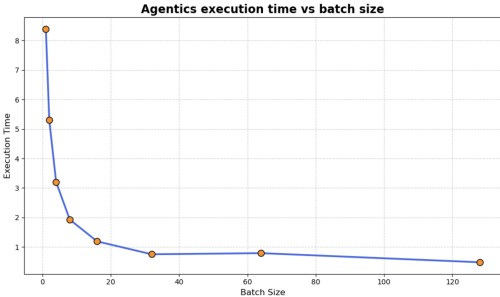
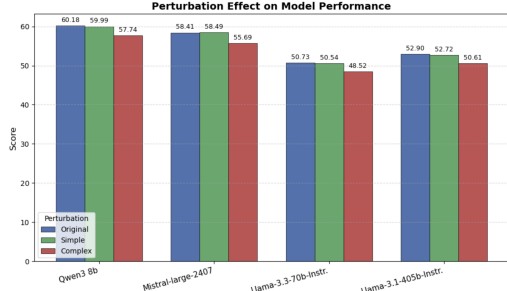

(a) Average time per question for `Qwen3-8B` across varying batch sizes.

(b) Performance remains high even after the perturbations with minimal drop.

Figure 3: Domain Specific MCQA Running Time and Perturbation Results.

## A.6 PROMPT OPTIMIZATION

Automatic prompt optimization (APO) is essential, as LLM performance is highly sensitive to prompt structure, tone, and formatting. The prompt function in Definition 9 plays a central role in logical transduction, which can be conceptualized as a negotiation of meaning between agents. Among various APO approaches (Ramnath et al., 2025; Li et al., 2025), logical transduction algebra naturally

supports OPRO-style methods (Yang et al., 2024; Ye et al., 2024; Opsahl-Ong et al., 2024), which follow a local search procedure based on a cycle of generate-select-evaluate. In the `Agentics` framework, the candidate prompt templates are generated by logical transduction from the meta-type that describes the prompt optimization task. Additionally, logical transduction algebra naturally parallelizes local search, such as generating and evaluating candidate prompts [5].

**Dataset**  We evaluated GSM8K (Cobbe et al., 2021) and FailureSensorIQ (Constantinides et al., 2025) benchmarks. The former enables direct comparison with existing APO techniques. For GSM8K, we use the first 500 training examples for prompt optimization and evaluate the final performance on the full test set. The latter benchmark allows us to assess additional gains beyond the default prompting provided by logical transduction. We randomize the dataset, selecting 500 examples for training and fixing 1,000 examples for the test set. In APO, the training set is used to construct demonstrations for candidate generation and to evaluate candidates during optimization.

**Methods**  We designed the meta-prompt for prompt generation by analyzing those from OPRO (Yang et al., 2024) and PE2 (Ye et al., 2024)[6]. Our focus is on the impact of local search hyperparameters on final test performance and the overall running time. Following Yang et al. (2024), the optimization meta-prompt includes 3 demonstration tasks and the top 8 candidates, along with their evaluated scores in ascending order. In our experiments, we vary the number of parallel candidate generations from 1 to 8, and the batch size for asynchronous LLM API calls from 1 to 20.

**Optimization Scope**  Following the common experiment settings (Yang et al., 2024; Ye et al., 2024; Zheng et al., 2025; Spiess et al., 2025), we optimize the prompt template for the zero-shot chain of thought technique (Kojima et al., 2022) by specifying the role, goal, expected output types, and the description of input task as well. This can be flexibly incorporated into APO, as candidate generation follows logical transduction from field descriptions to a prompt template. We initialize all prompt components as empty strings and iteratively refine them.

To understand the impact of prompt components, we manually constructed a prompt using the parameters and evaluated it with the LLaMA-3.3-70B model. The default prompt, which only shows the input and output field names, achieved just 5% accuracy. Adding an expected output description increased accuracy to 66%, and including all prompt parameters further improved it to 67%. These results suggest that optimizing both output expectations and imperative phrasing is essential for effective prompt design.

**Performance Improvement**  We present experiment results showing the improvement of test scores on both the GSM8K and MCQA datasets.

- The optimized prompt templates improve the test scores to 85 for Llama-3.3-70B and 91 for Qwen3-8B, which is consistent with findings reported in the literature. For the FailureSensorIQ dataset, the test score of Llama-3.3-70B was further improved to 54%.

- The plots with batch size 1 indicate that the Llama-3.3-70B model discovered better prompts when using a larger batch size. In contrast, the Qwen3-8B model identified a good prompt with a smaller batch size. However, it's important to note that the Qwen3 series models have been trained on data derived from the GSM8K dataset as well as other math-related datasets. This prior exposure may contribute to their relatively strong performance on GSM8K, even with smaller batch sizes or less prompt optimization effort, and should be considered when interpreting the results.

**Running Time**  Figure 5 shows the average running time per iteration during prompt optimization for the GSM8K amd MCQA dataset.

- In GSM8K, we observe that the improvement in running time is most significant at a batch size of 4, after which the gap gradually decreases. Beyond a batch size of 10, the running time saturates or increases due to the overhead caused by invalid outputs in the batch results.

---

[5]See Appendix for implementation details.

[6]Relevant open-source implementations are https://github.com/psunlpgroup/GreaterPrompt, https://github.com/stanfordnlp/dspy, https://github.com/google-deepmind/opro.

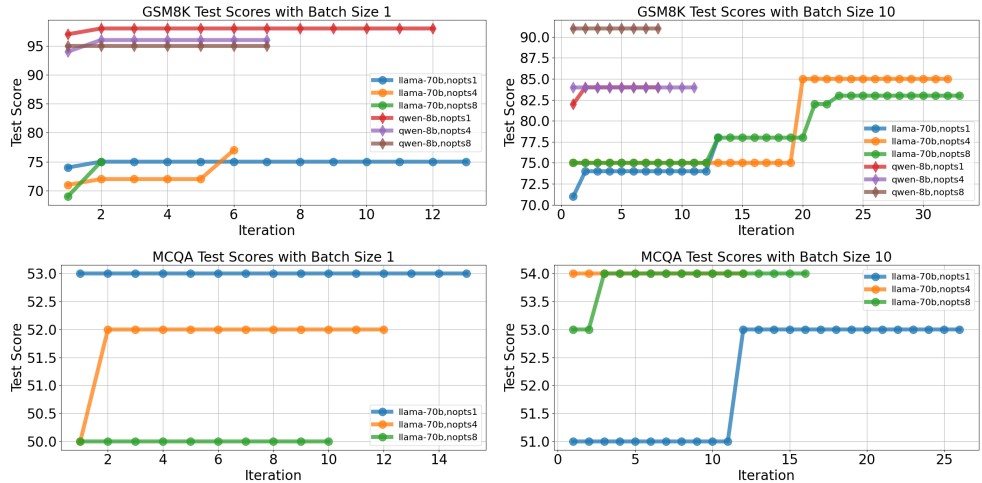

Figure 4: Improvement of test score over iterations: The x-axis represents the number of iterations, and the y-axis shows the test score evaluated using the best prompt template found up to that iteration.

- The running time results from the MCQA dataset follow similar trends to those observed with the GSM8K dataset. For smaller models like Qwen3-8B, if the model fails to follow instructions and produce output in the expected structured format, the transduction step often fails to return a valid JSON object. This leads to increased running time due to the additional overhead of the error recovery process.

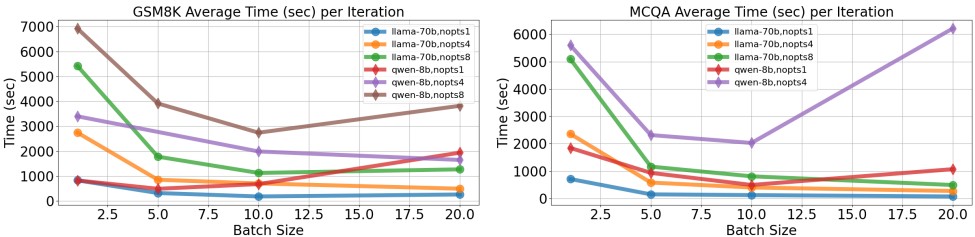

Figure 5: Average running time per iteration: The x-axis represents the batch size of the asynchronous execution, and the y-axis shows the average running time in seconds.

## B LOGICAL TRANSDUCTION ALGEBRA

### B.1 FORMALIZATION

We presented an abridged version of the formal Logical Transduction Algebra in the main paper. Our work is closely related to relational algebra (Codd, 1970) and the MapReduce programming model (Dean & Ghemawat, 2008). This enables the composition of data transformation pipelines and supports an efficient programming model that leverages the stateless and asynchronous nature of LLM inference.

#### B.1.1 ALGEBRAIC STRUCTURES

**Types and Agentic Structure**  We define types and meta-types, collectively referred to as the Agentic Structure ($AG$), and establish a sound algebra over the types and states within it.

**Definition 1** (Types).  Let $\Theta$ denote the universe of all possible *types*, $\Theta = \{X, Y, Z, T, \dots\}$, where each type $T \in \Theta$ is a collection of named fields $(s_i, T_{s_i})$:

$$T := \{(s_1, T_{s_1}), (s_2, T_{s_2}), \dots, (s_n, T_{s_n})\},$$

with each $s_i$ representing a string-valued slot name, and each $T_{s_i} \in \Theta$ denoting the corresponding type of that slot.

**Definition 2.** Given two types $X$ and $Y$, we define standard set operations component-wise:

$$
\begin{aligned}
X \cup Y &= \{(s_i, T_{s_i}) \mid (s_i, T_{s_i}) \in X \text{ or } (s_i, T_{s_i}) \in Y\}, \\
X \cap Y &= \{(s_i, T_{s_i}) \mid (s_i, T_{s_i}) \in X \text{ and } (s_i, T_{s_i}) \in Y\}, \\
X \setminus Y &= \{(s_i, T_{s_i}) \mid (s_i, T_{s_i}) \in X \text{ and } (s_i, T_{s_i}) \notin Y\}, \\
X \times Y &= \{((s_i, T_{s_i}), (s_j, T_{s_j})) \mid (s_i, T_{s_i}) \in X, (s_j, T_{s_j}) \in Y\}
\end{aligned}
$$

**Definition 3** (Agentic Structure $AG$)**.** An *Agentic structure $AG$* is a meta-type that bundles a type schema $s_{\text{atype}}$ [7] and a corresponding list of instances, referred to as **states** $s_{\text{states}}$:

$$
AG := \left\{ \begin{array}{l} s_{\text{atype}} : \Theta, \\ s_{\text{states}} : \text{List}[\Theta] \end{array} \right\}
$$

**Notation conventions:** Types are denoted by uppercase letters. Instances of types are denoted by lowercase letters, with $t : T$ indicating that $t$ is an instance of type $T$. Lists are written in boldface, so $\mathbf{t} : T$ represents a list of instances of type $T$. We use the shorthand $AG[X]$ to denote an Agentic structure with $s_{\text{atype}} = X$. A boldface lowercase symbol, such as $\mathbf{x} = AG[X]$, represents an instance of $AG[X]$. We also overload the notation to access the list of states: $x_i = \mathbf{x}[i] = \mathbf{x}.s_{\text{states}}[i]$ refers to the $i$-th state of the Agentic instance $\mathbf{x}$.

In Logical Transduction Algebra, we focus on structured data and its transformation around agents encapsulating LLMs. The algebraic structure of composing two Agentic instances of the same type can be shown as follows.

**Proposition 1** (Monoid of Agentic Instances)**.** Let $AG[X]$ be an Agentic structure and let $\xi$ be the set of all instances of $AG[X]$. Define a binary operation $\circ$ on $\xi$ such that for any $\mathbf{x}_1, \mathbf{x}_2 \in \xi$, their composition $\mathbf{x} = \mathbf{x}_1 \circ \mathbf{x}_2$ is an Agentic instance whose state list is the concatenation: $\mathbf{x}.s_{\text{states}} := \mathbf{x}_1.s_{\text{states}} \circ \mathbf{x}_2.s_{\text{states}}$. Then, the pair $(\xi, \circ)$ forms a *monoid*, where the identity element is the Agentic instance with an empty state list: $\mathbf{e}.s_{\text{states}} := []$.

*Proof.* We verify the three monoid properties:

*Closure:* Let $\mathbf{x}_1, \mathbf{x}_2 \in \xi$. Then $\mathbf{x} = \mathbf{x}_1 \circ \mathbf{x}_2$ has a state list formed by concatenating two valid state lists, which is itself valid. Hence, $\mathbf{x} \in \xi$.

*Associativity:* For any $\mathbf{x}_1, \mathbf{x}_2, \mathbf{x}_3 \in \xi$,

$$
\begin{aligned}
((\mathbf{x}_1 \circ \mathbf{x}_2) \circ \mathbf{x}_3).s_{\text{states}} &= (\mathbf{x}_1.s_{\text{states}} \circ \mathbf{x}_2.s_{\text{states}}) \circ \mathbf{x}_3.s_{\text{states}} \\
&= \mathbf{x}_1.s_{\text{states}} \circ (\mathbf{x}_2.s_{\text{states}} \circ \mathbf{x}_3.s_{\text{states}}) \\
&= (\mathbf{x}_1 \circ (\mathbf{x}_2 \circ \mathbf{x}_3)).s_{\text{states}}.
\end{aligned}
$$

*Identity:* Let $\mathbf{e} \in \xi$ be the Agentic instance with an empty state list. Then for any $\mathbf{x} \in \xi$,

$$
\begin{aligned}
(\mathbf{e} \circ \mathbf{x}).s_{\text{states}} &= [] \circ \mathbf{x}.s_{\text{states}} = \mathbf{x}.s_{\text{states}}, \\
(\mathbf{x} \circ \mathbf{e}).s_{\text{states}} &= \mathbf{x}.s_{\text{states}} \circ [] = \mathbf{x}.s_{\text{states}}.
\end{aligned}
$$

Thus, $(\xi, \circ)$ satisfies closure, associativity, and identity, and is therefore a monoid. $\square$

The standard operators follow standard algebraic principles such as the product, $\mathbf{x} \times \mathbf{y}$ for $\mathbf{x} \in AG[X]$ and $\mathbf{y} \in AG[Y]$, the equivalence, $\mathbf{x}_1 \sim \mathbf{x}_2$ for $\mathbf{x}_1, \mathbf{x}_2 \in AG[X]$, and the quotient, $\mathbf{z}/\mathbf{y}$ for $\mathbf{z} \in AG[X \times Y]$ and $\mathbf{y} \in AG[Y]$.

---

[7]Given two types $X$ and $Y$, the standard set operations such as union, intersection, complement, and product can be defined component-wise.

**Product of the Agentic Structure** Next, we define the product of Agentic structures, a construction that plays a foundational role in modeling and executing complex, multi-dimensional data workflows. By combining two Agentic structures into a single product structure, we can represent composite types—such as paired entities, coupled processes, or input-output relationships—within a unified algebraic framework. This formulation ensures that operations applied to states remain well-defined, type-safe, and composable, preserving the monoidal properties of each component. The product structure is especially valuable in scenarios involving joint reasoning, parallel transformations, or structured transductions across heterogeneous data streams.

**Definition 4** (Product of Agentic Structures). Let $AG[X]$ and $AG[Y]$ be two Agentic structures defined over distinct types $X$ and $Y$, respectively. We define their product as a new Agentic structure $AG[T]$, where the type $T$ is the Cartesian product of the two types:

$$T : X \times Y.$$

Given instances $\mathbf{x} : AG[X]$ and $\mathbf{y} : AG[Y]$, we define an instance $\mathbf{t} : AG[T]$ such that:

$$\mathbf{t}.s_{\text{states}} = (\mathbf{x}.s_{\text{states}}, \mathbf{y}.s_{\text{states}}),$$

i.e., the state list of $\mathbf{t}$ is the pair of state lists from $\mathbf{x}$ and $\mathbf{y}$.

**Proposition 2** (Monoid of Agentic Product). Let $\xi_X$ and $\xi_Y$ be the set of all instances of $AG[X]$ and $AG[Y]$, respectively, and let $\xi_T$ be the set of all instances of $AG[T]$.

Define a binary operation $\circ$ on $\xi_T$ as follows:

$$(\mathbf{x}_1, \mathbf{y}_1) \circ (\mathbf{x}_2, \mathbf{y}_2) := (\mathbf{x}_1 \circ \mathbf{x}_2, \mathbf{y}_1 \circ \mathbf{y}_2),$$

where $\circ$ on each component denotes concatenation of state lists:

$$(\mathbf{x}_1 \circ \mathbf{x}_2).s_{\text{states}} := \mathbf{x}_1.s_{\text{states}} \circ \mathbf{x}_2.s_{\text{states}},$$

and similarly for $\mathbf{y}_1 \circ \mathbf{y}_2$.

Then, the structure $(\xi_T, \circ)$ forms a *monoid*, with the identity element given by the pair of Agentic instances with empty state lists:

$$\mathbf{e}_T := (\mathbf{e}_X, \mathbf{e}_Y), \quad \text{where } \mathbf{e}_X.s_{\text{states}} = [], \quad \mathbf{e}_Y.s_{\text{states}} = [].$$

*Proof.* We verify the three monoid properties for $(\xi_T, \circ)$.

*Closure:* Let $(\mathbf{x}_1, \mathbf{y}_1), (\mathbf{x}_2, \mathbf{y}_2) \in \xi_T$. Then their composition is:

$$(\mathbf{x}_1 \circ \mathbf{x}_2, \mathbf{y}_1 \circ \mathbf{y}_2),$$

where each component is a valid Agentic instance due to closure in $(\xi_X, \circ)$ and $(\xi_Y, \circ)$. Hence, the result is in $\xi_T$.

*Associativity:* Let $(\mathbf{x}_1, \mathbf{y}_1), (\mathbf{x}_2, \mathbf{y}_2), (\mathbf{x}_3, \mathbf{y}_3) \in \xi_T$. Then:

$$\begin{aligned}
((\mathbf{x}_1, \mathbf{y}_1) \circ (\mathbf{x}_2, \mathbf{y}_2)) \circ (\mathbf{x}_3, \mathbf{y}_3) &= (\mathbf{x}_1 \circ \mathbf{x}_2, \mathbf{y}_1 \circ \mathbf{y}_2) \circ (\mathbf{x}_3, \mathbf{y}_3) \\
&= ((\mathbf{x}_1 \circ \mathbf{x}_2) \circ \mathbf{x}_3, (\mathbf{y}_1 \circ \mathbf{y}_2) \circ \mathbf{y}_3) \\
&= (\mathbf{x}_1 \circ (\mathbf{x}_2 \circ \mathbf{x}_3), \mathbf{y}_1 \circ (\mathbf{y}_2 \circ \mathbf{y}_3)) \\
&= (\mathbf{x}_1, \mathbf{y}_1) \circ ((\mathbf{x}_2, \mathbf{y}_2) \circ (\mathbf{x}_3, \mathbf{y}_3)),
\end{aligned}$$

using associativity in each component.

*Identity:* Let $\mathbf{e}_T := (\mathbf{e}_X, \mathbf{e}_Y)$, where $\mathbf{e}_X$ and $\mathbf{e}_Y$ are identity elements in $\xi_X$ and $\xi_Y$, respectively. Then for any $(\mathbf{x}, \mathbf{y}) \in \xi_T$:

$$\begin{aligned}
(\mathbf{e}_X, \mathbf{e}_Y) \circ (\mathbf{x}, \mathbf{y}) &= (\mathbf{e}_X \circ \mathbf{x}, \mathbf{e}_Y \circ \mathbf{y}) = (\mathbf{x}, \mathbf{y}), \\
(\mathbf{x}, \mathbf{y}) \circ (\mathbf{e}_X, \mathbf{e}_Y) &= (\mathbf{x} \circ \mathbf{e}_X, \mathbf{y} \circ \mathbf{e}_Y) = (\mathbf{x}, \mathbf{y}).
\end{aligned}$$

Hence, $\mathbf{e}_T$ is the identity element. $\qquad\square$

**Quotient of the Agentic Structure**    To complement the expressiveness of product structures, the quotient of Agentic structures provides a principled mechanism for abstraction and generalization. By defining an equivalence relation over Agentic instances—such as grouping together states that differ only in irrelevant or redundant dimensions—we can collapse fine-grained distinctions into coarser, semantically meaningful categories. This is especially useful in scenarios involving behavioral equivalence, or clustering of similar agentic behaviors. The quotient structure enables reasoning at a higher level of abstraction while preserving the algebraic properties of the original system. In distributed settings, it supports compression, deduplication, and aggregation of stateful computations.

**Definition 5** (Equivalence Relation on Agentic Instances). Let $\xi_X$ be the set of Agentic instances over type $X$.

An equivalence relation $\sim$ on $\xi_X$ is defined by a relation $\mathcal{R}$ on state lists $\mathbf{s} : X$ such that for any $\mathbf{x}, \mathbf{y} \in \xi_X$,

$$\mathbf{x} \sim \mathbf{y} \iff \mathcal{R}(\mathbf{x}.s_{\text{states}}, \mathbf{y}.s_{\text{states}}),$$

where $\mathcal{R}$ satisfies the following properties:

- *Reflexivity:* $\mathcal{R}(\mathbf{s}, \mathbf{s})$ for all state lists $\mathbf{s}$.

- *Symmetry:* If $\mathcal{R}(\mathbf{s}_1, \mathbf{s}_2)$, then $\mathcal{R}(\mathbf{s}_2, \mathbf{s}_1)$.

- *Transitivity:* If $\mathcal{R}(\mathbf{s}_1, \mathbf{s}_2)$ and $\mathcal{R}(\mathbf{s}_2, \mathbf{s}_3)$, then $\mathcal{R}(\mathbf{s}_1, \mathbf{s}_3)$.

The specific form of $\mathcal{R}$ depends on the semantics of the Agentic structure. A common choice is *statewise equivalence*, defined below.

**Definition 6** (Statewise Equivalence of Agentic Instances). Let $\xi_X$ be the set of Agentic instances over type $X$.

Define an equivalence relation $\sim$ on $\xi_X$ such that for any $\mathbf{x}, \mathbf{y} \in \xi_X$,

$$\mathbf{x} \sim \mathbf{y} \iff \mathbf{x}.s_{\text{states}} \equiv \mathbf{y}.s_{\text{states}},$$

where $\equiv$ denotes elementwise equivalence of state lists. That is,

$$\mathbf{x}.s_{\text{states}} = [x_1, x_2, \ldots, x_n], \quad \mathbf{y}.s_{\text{states}} = [y_1, y_2, \ldots, y_n],$$

and for all $i = 1, \ldots, n$, we have $x_i \approx y_i$ under a given equivalence relation $\approx$ on $X$.

The relation $\approx$ on $X$ may be defined in various ways, such as:

- *Syntactic equality:* $x_i = y_i$.

- *Observational equivalence:* $f(x_i) = f(y_i)$ for some observable function $f : X \to O$.

- *Abstract equivalence:* $x_i$ and $y_i$ belong to the same equivalence class under a domain-specific partition of $X$.

This relation groups Agentic instances whose state trajectories are equivalent up to the equivalence of individual states.

**Definition 7** (Quotient of Agentic Structure). Let $AG[X]$ be an Agentic structure over type $X$, and let $\sim$ be an equivalence relation on the set of Agentic instances $\xi_X$.

The *quotient Agentic structure*, denoted $AG[X/\sim]$, is defined as follows:

- The type $X/\sim$ is the set of equivalence classes of $X$ under the induced relation $\approx$ on individual states.

- The set of instances $\xi_{X/\sim}$ consists of equivalence classes $\langle \mathbf{x} \rangle$ of Agentic instances $\mathbf{x} \in \xi_X$, where

$$\langle \mathbf{x} \rangle := \{ \mathbf{y} \in \xi_X \mid \mathbf{y} \sim \mathbf{x} \}.$$

- The state list of an equivalence class $\langle \mathbf{x} \rangle$ is defined as:

$$\langle \mathbf{x} \rangle.s_{\text{states}} := \{ \mathbf{y}.s_{\text{states}} \mid \mathbf{y} \sim \mathbf{x} \}.$$

This structure abstracts over individual Agentic instances by identifying those whose states are equivalent.

**Proposition 3** (Monoid Structure on Quotient Agentic Structure). Let $(\xi_X, \circ)$ be a monoid of Agentic instances over type $X$, and let $\sim$ be a congruence relation on $\xi_X$, i.e., for all $\mathbf{x}_1 \sim \mathbf{x}_2$ and $\mathbf{y}_1 \sim \mathbf{y}_2$, we have:

$$\mathbf{x}_1 \circ \mathbf{y}_1 \sim \mathbf{x}_2 \circ \mathbf{y}_2.$$

Then, the quotient structure $(\xi_X / \sim, \circ)$ forms a monoid, where:

- Elements are equivalence classes $\langle \mathbf{x} \rangle$.

- The operation is defined by:
$$\langle \mathbf{x} \rangle \circ \langle \mathbf{y} \rangle := \langle \mathbf{x} \circ \mathbf{y} \rangle.$$

- The identity is $\langle \mathbf{e} \rangle$, where $\mathbf{e}$ is the identity in $(\xi_X, \circ)$.

*Proof.* We verify the monoid properties on the quotient structure:

*Well-definedness:* If $\mathbf{x}_1 \sim \mathbf{x}_2$ and $\mathbf{y}_1 \sim \mathbf{y}_2$, then by congruence:
$$\mathbf{x}_1 \circ \mathbf{y}_1 \sim \mathbf{x}_2 \circ \mathbf{y}_2,$$
so $\langle \mathbf{x}_1 \circ \mathbf{y}_1 \rangle = \langle \mathbf{x}_2 \circ \mathbf{y}_2 \rangle$.

*Associativity:* Follows from associativity in $\xi_X$:
$$\langle \mathbf{x} \rangle \circ (\langle \mathbf{y} \rangle \circ \langle \mathbf{z} \rangle) = \langle \mathbf{x} \circ (\mathbf{y} \circ \mathbf{z}) \rangle = \langle (\mathbf{x} \circ \mathbf{y}) \circ \mathbf{z} \rangle = (\langle \mathbf{x} \rangle \circ \langle \mathbf{y} \rangle) \circ \langle \mathbf{z} \rangle.$$

*Identity:* For any $\langle \mathbf{x} \rangle$,
$$\langle \mathbf{e} \rangle \circ \langle \mathbf{x} \rangle = \langle \mathbf{e} \circ \mathbf{x} \rangle = \langle \mathbf{x} \rangle, \quad \langle \mathbf{x} \rangle \circ \langle \mathbf{e} \rangle = \langle \mathbf{x} \circ \mathbf{e} \rangle = \langle \mathbf{x} \rangle.$$

$\square$

**Example 1** (Quotient of a Product Agentic Structure). Let $AG[X]$ and $AG[Y]$ be Agentic structures over types $X$ and $Y$, respectively. Their product $AG[T]$ is defined by
$$s_{\text{atype}} = X \times Y, \quad s_{\text{states}} = (\mathbf{x}.s_{\text{states}}, \mathbf{y}.s_{\text{states}})$$
for instances $\mathbf{x} \in AG[X]$ and $\mathbf{y} \in AG[Y]$.

We define an equivalence relation $\sim$ on $\xi_T$ such that for any $\mathbf{t}_1, \mathbf{t}_2 \in \xi_T$,
$$\mathbf{t}_1 \sim \mathbf{t}_2 \iff \forall i, \; x_i^{(1)} \approx x_i^{(2)},$$
where $x_i^{(1)}$ and $x_i^{(2)}$ are the first components of the $i$-th state in $\mathbf{t}_1$ and $\mathbf{t}_2$, respectively, and $\approx$ is an equivalence relation on $X$.

As a concrete example, let the types and the equivalence $\approx$ be defined as:

- $X = \{\text{Red}, \text{Green}, \text{Blue}\}$ (colors),

- $Y = \{\text{Circle}, \text{Square}\}$ (shapes),

- Define $\approx$ on $X$ by:
$$\text{Red} \approx \text{Green}, \quad \text{Blue} \not\approx \text{Red}, \quad \text{Blue} \not\approx \text{Green}.$$

Consider two Agentic instances:
$$\mathbf{t}_1.s_{\text{states}} = [(\text{Red}, \text{Circle}), (\text{Green}, \text{Square})],$$
$$\mathbf{t}_2.s_{\text{states}} = [(\text{Green}, \text{Circle}), (\text{Red}, \text{Square})].$$

Then $\mathbf{t}_1 \sim \mathbf{t}_2$ because:
$$\text{Red} \approx \text{Green}, \quad \text{Green} \approx \text{Red}.$$
Note that the shape components (second elements) are not constrained by the equivalence relation.

The quotient Agentic structure $AG[T / \sim]$ consists of equivalence classes $\langle \mathbf{t} \rangle$ of Agentic instances under $\sim$, where:
$$\langle \mathbf{t} \rangle := \{\mathbf{t}' \in \xi_T \mid \mathbf{t}' \sim \mathbf{t}\}.$$

This structure abstracts over differences in the first component of the state tuples according to $\approx$, while preserving the full state list structure.

### B.1.2 THE TRANSDUCTION OPERATOR

Equipped with Agentic structures that form a monoid, we obtain a sound abstraction for composing the data workflows in a functional programming style. This foundation enables the introduction of the *logical transduction operator*, which utilizes LLMs as transductive inference engines. We now define a series of logical transduction operators, organized by increasing levels of complexity. These operators are designed with explicit consideration of types and Agentic structures.

**Transduction Operator Overloading**

**Definition 8** (Transduction). Given an information object $x$ and a target Agentic structure $AG[Y]$, the *transduction* of $x$ into $AG[Y]$ is defined as:

$$\mathbf{y} := Y \ll x = \bigcup_i (y_i, T_{s_i}),$$

where each $y_i \in T_{s_i}$ is a value assigned to slot $s_i$ of type $T_{s_i}$, logically inferred from $x$. Here, the operator $\ll$ denotes a logical transduction process, implemented via an LLM, that maps $x$ to a structured output conforming to the type $Y$.

To support the definition of logical transduction operators, we introduce a generic function that renders typed objects into textual representations suitable for LLM input.

**Definition 9** (Prompt Function). Given a type $T \in \Theta$, a *prompt function* $P$ is a mapping that renders a list of states $\mathbf{t} : T$ into an information object, leveraging the string-valued slot names associated with $T$. Formally, $P : \text{List}[T] \to \texttt{str}$.

Prompt functions serve as a bridge between structured data and natural language, enabling logical transduction operators to interface with LLMs by converting typed instances into semantically meaningful prompts.

We now define two specific forms of logical transduction: zero-shot and few-shot.

**Definition 10** (Zero-Shot Logical Transduction). Let $\mathbf{x} = AG[X]$ and $\mathbf{y} = AG[Y]$ be Agentic structures over types $X$ and $Y$, respectively. A *zero-shot logical transduction* from $\mathbf{x}$ to $\mathbf{y}$ is defined component-wise as:

$$\mathbf{y}[i] = Y \ll P(\mathbf{x}[i]),$$

where $P : X \to \texttt{str}$ is a prompt function that renders each instance $\mathbf{x}[i]$ into a textual prompt.

Next, we also show more overloaded transduction operators in the case of the few-shot transduction, $\mathbf{y}[i] = Y \ll (P(\mathbf{x}[i]) \oplus FS(\mathbf{x}, \mathbf{y}))$ with a few-shot function $FS(\mathbf{x}, \mathbf{y}) := P((\mathbf{x}', \mathbf{y}'))$, and a syntactic sugar such as self-transduction.

**Definition 11** (Few-Shot Logical Transduction). Let $\mathbf{x} = AG[X]$ and $\mathbf{y} = AG[Y]$ be Agentic structures over types $X$ and $Y$, respectively. A *few-shot logical transduction* from $\mathbf{x}$ to $\mathbf{y}$ is defined for all indices $i$ such that $\mathbf{y}[i] = \emptyset$ as:

$$\mathbf{y}[i] = Y \ll (P(\mathbf{x}[i]) \oplus FS(\mathbf{x}, \mathbf{y})),$$

where:

- $P : X \to \texttt{str}$ is a prompt function that renders an instance $\mathbf{x}[i]$ into a textual prompt.

- $\oplus$ denotes prompt concatenation.

- $FS(\mathbf{x}, \mathbf{y})$ is the *few-shot context*, defined as:
  $$FS(\mathbf{x}, \mathbf{y}) := P((\mathbf{x}', \mathbf{y}')),$$
  where $(\mathbf{x}', \mathbf{y}')$ is the projection of $(\mathbf{x}, \mathbf{y})$ onto the subset of indices for which $\mathbf{y}[i] \neq \emptyset$.

Finally, we introduce self-transduction as syntactic sugar within the programming model for logical transductions.

**Definition 12** (Self Transduction). Let $\mathbf{x} \in AG[X]$ be an Agentic structure, and let $Y, Z \subset X$ be two disjoint subsets of types. A *self transduction* is a function that produces a modified Agentic structure $\mathbf{x}' \in AG[X]$, defined as:

$$\mathbf{x}' = \mathbf{x} \ll_{Y,Z} := \mathbf{x} \cup (\mathbf{x}[Y] \ll \mathbf{x}[Z]),$$

where $\mathbf{x}[Y]$ denotes the *rebind operator*, which extracts an $AG[Y]$ from $\mathbf{x}$ by retaining only the slots in $Y$ that overlap with $X$.

**Properties of Transduction Operator**  Next, we formalize key properties of the transduction operator. These properties are foundational for enabling scalable, parallel, and composable computation.

**Proposition 4** (Conditional Determinism). Let $\sigma$ denote a fixed transduction context, which may include components such as a few-shot context, additional instructions, external tools, or memory. Let the LLM configuration—comprising model weights, temperature, and decoding strategy—also be fixed. Then, for any input $x_i$, the transduction $y_i := Y \ll x_i$ is deterministic.

*Proof.* When the model parameters and transduction context $\sigma$ are fixed, and the LLM is configured with deterministic settings (e.g., temperature set to zero and caching enabled), the output $y_i$ is uniquely determined by the input $x_i$ and the context $\sigma$. Therefore, the transduction process is deterministic under these conditions. □

**Proposition 5** (Statelessness). Logical transduction operators are stateless. The output of a transduction $y_i := Y \ll x_i$ depends only on $x_i$ and the transduction context, and not on any prior or future transductions.

*Proof.* By definition, the transduction operator $\ll$ does not rely on conversational memory or sequential state. Each $y_i$ is computed independently from $x_i$ and $\sigma$, enabling asynchronous evaluation. □

**Proposition 6** (Compositionality). Let $AG[X]$, $AG[Y]$, and $AG[Z]$ be Agentic structures over types $X$, $Y$, and $Z$, respectively. Suppose $\mathbf{y}' = \mathbf{y} \ll \mathbf{x}$, and $\mathbf{z}' = \mathbf{z} \ll \mathbf{y}'$. Then the composite transduction holds, $\mathbf{z}' = \mathbf{z} \ll \mathbf{y} \ll \mathbf{x}$.

*Proof.* Each transduction step applies $\ll$ component-wise to the state list of the input Agentic structure. Since the output of $\mathbf{y} \ll \mathbf{x}$ is an Agentic structure $AG[Y]$, it can be used as input to the next transduction. Thus, the composition is well-defined and yields $\mathbf{z}'$. □

**Implications for Distributed and Concurrent Computing**  These properties make the transduction operator $\ll$ particularly well-suited for distributed and concurrent computing paradigms. The *Conditional Determinism* ensures reproducibility and traceability in distributed pipelines, the *Statelessness* enables parallel execution, allowing transductions to be mapped across shards of data without coordination or shared state, and the *Compositionality* supports modular pipeline construction, akin to functional composition in MapReduce, where intermediate Agentic structures can be chained and reused.

### B.1.3 ASYNCHRONOUS MAPREDUCE

The programming model of the `Agentics` supports asynchronous execution of mapping and reduction operations over Agentic structures, enabling scalable and composable data workflows. We formalize these operations as `aMap` and `aReduce`, which extend the MapReduce by Dean & Ghemawat (2008).

**Definition 13** (Asynchronous Map (`aMap`)). Let $AG[X]$ be an Agentic structure over type $X$, and let $f : X \to \text{List}[Y]$ be an asynchronous mapping function. Then the asynchronous map operator is defined as:

$$\text{aMap} : (AG[X], f) \to AG[Y],$$

where the output Agentic structure $\mathbf{y} = \text{aMap}(\mathbf{x}, f)$ satisfies: $\mathbf{y}.s_{\text{states}} = \bigcup_i f(x_i)$, and the union preserving the original order of inputs.

The function $f$ may return an empty list by removing $x_i$ from the output acting as a filter, map each $x_i$ to a single output acting as a transformer, or map each $x_i$ to multiple outputs acting as fan-out. Note that `aMap` operator is executed asynchronously across all input states, enabling parallelism and scalability in distributed environments.

The `aMap` operator is executed asynchronously across all input states, enabling parallelism and scalability in distributed environments.

**Definition 14** (Asynchronous Reduce (`aReduce`))**.** Let $AG[X]$ be an Agentic structure and let $f : \mathrm{List}[X] \to Y$ be an asynchronous reduction function. Then the asynchronous reduce operator is defined as:

$$\mathtt{aReduce} : (AG[X], f) \to AG[Y],$$

where the output Agentic structure $\mathbf{y} = \mathtt{aReduce}(\mathbf{x}, f)$ satisfies: $\mathbf{y}.s_{\mathrm{states}} = f(\mathbf{x})$.

Unlike `aMap`, which applies $f$ to each state individually, `aReduce` applies $f$ to the entire states $\mathbf{x}$ at once. This is useful for summarization or aggregation, such as generating a report or computing statistics over the full dataset. Since LLMs have limited context windows, applying `aReduce` to a large dataset may be intractable. In such cases, scalable strategies such as hierarchical or batched reduction can be employed by applying `aReduce` to random subsets and merging the results.

**Composability with Logical Transduction** `aMap` and `aReduce` can be composed with the logical transduction operator $\ll$ to build expressive and modular workflows. For example,

$$\mathbf{y} = \mathtt{aMap}(\mathbf{x}, \, x \mapsto Y \ll x), \quad \mathbf{z} = \mathtt{aReduce}(\mathbf{y}, \, f),$$

where the transduction $Y \ll x$ is embedded within the mapping function. As we can see, Agentic structure enables distributed, asynchronous, and semantically typed computation over structured data.

## C  Design Patterns and Use Cases

The `Agentics` framework provides a concrete realization of the logical transduction algebra. We elaborate on various design patterns and use cases for domain-specific multi-choice QA, text-to-SQL pipelines, clustering, and automatic prompt optimization. This section demonstrates how the `Agentics` framework supports a wide range of generative structured data workflows through reusable design patterns. Each use case highlights a different aspect of the `Agentics` programming model, showcasing its flexibility, scalability, and composability.

### C.1  Semantic Parsing Text-to-SQL

Text-to-SQL is an essential task for broadening the accessibility of structured data interaction, allowing users to query databases without needing to understand the underlying decisions made by data engineers. Loosely considered a translation task, questions are posed to a database and first translated into SQL queries before being executed and answers retrieved and answers retrieved. In practice, this task involves multiple stages of reasoning, whereby one has to interact with the schema of the structured data, as well as understand and decompose the question into its constituent parts.

**Data Models** `Agentics` supports this workflow by chaining multiple transduction steps and integrating them with traditional Python logic. First, let's consider setting up the Pydantic types with the required task information.

```python
class Text2SQLTask(BaseModel):
    question: str = Field(description="The input natural language question.")
    ddl: str = Field(description="The database schema in DDL format (e.g., CREATE TABLE
        statements).")
    sql_query: Optional[str] = Field(description="The SQL query to be generated from the
        question.")
    execution_result: Optional[List[Dict[str, str]]] = Field(description="The resulting table
        from executing the SQL query.")
```

Listing 2: Data model for the simplified Text2SQL task

From the above Pydantic types, the simplified task is to map the question and DDL to the sql_query. However, we can break down this complex operation into declarative data modeling steps to improve task performance, including:

- enriching the database so that there are additional fields like description of the schema and business descriptions

- decompose the user question into constituent parts input that can be also a part of optimization!

- optimize the prompt template using the final input fields.

```python
class Text2SQLTask(BaseModel):
    question: str = Field(description="The input natural language question.")
    ddl: str = Field(description="The database schema in DDL format (e.g., CREATE TABLE
        statements).")
    enrichment: Optional[DB] = Field(description="Additional database enrichments  that
        applies to all problem instances")
    sql_query: Optional[str] = Field(description="The SQL query to be generated from the
        question.")
    execution_result: Optional[List[Dict[str, str]]] = Field(description="The resulting table
        from executing the SQL query.")

class DB(BaseModel):
    description: Optional[str] = Field(description="A Description of the business purpose of
        the db, what use cases it is good for how what type of information it contain")
    keywords: Optional[list[str]] = Field(description="A list of keywords describing the
        content of the database. Produce Keywords that are: Domain-Relevant: Reflects the
        thematic area (e.g., education, healthcare, finance). Purpose-Oriented: Indicates the
         type of insights the database supports (e.g., performance tracking, demographic
        analysis). Unambiguous: Avoids generic or overly broad terms. Interoperable: Aligns
        with standard taxonomies when possible (e.g., DataCite or UNSDG classification).
        Examples of Strong Keywords: student_outcomes, climate_metrics, financial_forecasting
        , public_health_indicators, supply_chain_kpis")
    few_shots: Optional[list[QuestionSQLPair]] = Field(description="A selection of the
        generated question-sql pair to be used as examples of how to generate a sql from a
        question.")
    subquestions: Optional[list[str]] = Field(description="a list of subquestions inside
        question")
    schema_link: Optional[str] = Field(description="the output of the schema linker in the
        form of DDL script showing relevant table, column, and values for the question")

class QuestionSQLPair(BaseModel):
    question: Optional[str]
    sql_query: Optional[str]
```

Listing 3: Data model for the compositional Text2SQL task

**Meta-Prompts and Prompt Templates**  We define the initial prompt that performs the main text-to-SQL task. The following shows the prompt template used as input to the experiment, which may be optionally modified by the automated prompt optimization flag.

```
prompt_template = "
Translate the input natural language **question** to a valid SQLite query that can be executed
    on the following  database in **dbs**.
Do your best to apply the following rules when generating SQL.
- Cleary understand the **question** and given database description in **dbs**.
- Database description is given in the **dbs** with the following fields.
- The database description under **dbs** contains DDL scripts or natural language description
    of the database, tables, columns, and values.
- Only SELECT statements are allowed, do not produce any DDL or DML.
- When writing `SELECT <column>`, only include the columns specifically mentioned in the
    question.
- Use **evidence** to find correct column names or the values of the columns or other
    expressions.
- If you see 'None' or 'None' in the [Value examples] for a column, prioritize using `JOIN <
    table>` or `WHERE <column> IS NOT NULL` to handle potential missing data effectively.
- Use `WHERE <column> IS NOT NULL` in `WHERE` if you are sorting with `<column>`.
- Use alias in `SELECT` is consistently in the expressions.
- Use `WHERE <alias> IS NOT NULL` in `WHERE` if you are sorting with `<alias>`.

Input is provided under SOURCE.
"{input_spec_str}"

Generate Output as JSON decodable format
"{{"
generated_sql_query: a valid SQLite query that translates nautral language question
"}}"\n
"
```

Listing 4: The Prompt Template and Meta-Prompt for Automatic Prompt Optimization

**Main Algorithm**  Next, we show how to define the compositional text-to-SQL pipeline in `Agentics`. We see that the entire pipeline can be implemented through compositions of logical transductions defined over the data models.

```
text2sql("enrichment.keywords") << text2sql("enrichment.description") << text2sql("ddl_schema"
    )
text2sql("few_shots.question", "few_shots.sql_query") << text2sql("ddl_schema", "enrichment.
    description")  # synthetic pair
text2sql("few_shots") = text2sql("few_shots") + k_shot  # additional augmentations
```

```
4  text2sql("few_shots") = text2sql.filter(valid_sql, "few_shots")  # executes sql
5  text2sql("enrichment.subquestions") << text2sql("question")
6  text2sql("enrichment.linked_schema") << text2sql("enrichment.ddl_schema", "enrichment.
       subquestions", "question")
7  input_fields = ["question", "enrichment"]
8  text2sql.instructions = optimize(prompt_template, input_fields)
9  text2sql("sql_query") << text2sql(*input_fields)
10 text2sql("execution_result") = text2sql.amap(execute_sql_query)
```

Listing 5: Pseudo-code for the Compositional Text-to-SQL Workflow

## C.2 DOMAIN-SPECIFIC MULTIPLE CHOICE QUESTION ANSWERING

Domain-specific Multiple Choice Question Answering (MCQA) tasks present unique challenges for LLMs, particularly when grounded in technical domains that are unfamiliar or underrepresented in pretraining corpora.

**FailureSensorIQ Benchmark**    In this subsection, we demonstrate how the `Agentics` framework supports structured reasoning and robust performance on the *FailureSensorIQ* benchmark—a dataset designed to evaluate LLMs' understanding of failure modes and sensor relationships in Industry 4.0 (Constantinides et al., 2025).

Unlike widely used QA datasets such as MMLU (Hendrycks et al., 2021) or MedQA Li et al. (2020), FailureSensorIQ introduces a novel domain with no prior exposure to the models under evaluation. This makes it a strong testbed for assessing generalization and reasoning capabilities in high-stakes, real-world industrial contexts. The benchmark includes 8,296 questions across 10 assets, with both single- and multi-answer formats. Despite the presence of strong reasoning models, the best-performing `openai-o1` achieves only 60.4 precent accuracy on single-answer questions, underscoring the dataset's difficulty.

**Schema-Guided LLM Reasoning**    The `Agentics` approach to MCQA leverages self-transduction and schema-driven prompting using `Pydantic` models. This structured prompt format contrasts with the loosely formatted natural language prompts used in baseline evaluations. By explicitly encoding the input-output schema (e.g., JSON fields for question, options, and selected answers), `Agentics` reduces decoding errors and enforces type safety. This is particularly beneficial in multi-answer settings, where ambiguity in output formatting can lead to evaluation mismatches and degraded performance.

We observe that this structured prompting pattern not only improves accuracy but also enhances robustness to perturbations. For example, when question phrasing or distractor options are altered, schema-constrained decoding helps maintain consistent model behavior. This suggests that `Agentics`' structured approach offers a degree of perturbation resilience, addressing one of the key weaknesses identified in the original FailureSensorIQ benchmark.

In addition to accuracy improvements, the `Agentics` framework enables parallel batch execution of MCQA tasks, significantly reducing inference time. This is achieved through the aMap operation, which distributes structured prompts across multiple model invocations. Compared to sequential prompting, this design pattern yields substantial speedups, making it practical for large-scale evaluation and deployment.

**Data Model**    The data model for domain-specific MCQA in `Agentics` is defined using `Pydantic`, which enforces structural constraints and type safety during both prompt construction and response decoding. This schema-guided approach ensures that the model's outputs conform to expected formats, reducing parsing errors and improving evaluation reliability.

The `FailureSensorIQ` class encapsulates the core components of each QA instance, including the question text, list of options, associated metadata (e.g., asset name, question type), and the model-generated answer. The nested `Answer` class captures the model's selected answer, a numerical confidence score, and a free-text explanation that provides insight into the model's reasoning process.

```
1  class Answer(BaseModel):
2      answer_letter: str = Field(description="The selected answer letter")
3      confidence: float = Field(description="Confidence score")
4      assessment: str = Field(description="Rationale for the answer")
```

```python
class FailureSensorIQ(BaseModel):
    id: int = Field(description="Unique identifier for the question")
    question: str = Field(description="The question text")
    options: List[str] = Field(description="List of answer options")
    option_ids: List[str] = Field(description="List of option identifiers")
    asset_name: str = Field(description="Name of the industrial asset")
    relevancy: str = Field(description="Relevancy context or metadata")
    question_type: str = Field(description="Type of question")
    subject: str = Field(description="Subject or topic of the question")
    system_answer: Answer = Field(description="Model-generated answer object")
```

Listing 6: Data Model for Domain-Specific Multiple Choice QA

**Main Algorithm** The workflow begins by instantiating an AG object with the `FailureSensorIQ` schema and a specified batch size. Each example from the dataset is parsed into a structured `FailureSensorIQ` instance and appended to the agent's internal state.

The core operation is the self transduction, which performs schema-guided inference over the specified input fields—such as question, options, and asset name—and generates structured outputs in the system answer field. The transduction is guided by a natural language instruction that defines the task: selecting the most plausible answer from a set of options, along with a confidence score and rationale.

```python
# Initialize the \texttt{Agentics} benchmark with the FailureSensorIQ schema and batch size
fsiq_benchmark = AG(atype=FailureSensorIQ, batch_size=40)

# Load dataset and populate agent states
dataset = load_dataset("cc4718/FailureSensorIQ")
for example in dataset:
    fsiq_benchmark.states.append(FailureSensorIQ(**example))

# Run self-transduction with structured input and output fields
fsiq_benchmark = await fsiq_benchmark.self_transduction(
    input_fields=[
        "question", "options", "option_ids",
        "asset_name", "relevancy", "question_type", "subject"
    ],
    output_fields=["system_answer"],
    instructions=(
        "Read the input questions, all possible answers, and background task information. "
        "This is a multiple choice test, where one of the options is true and the others are
            false. "
        "Select the answer with the highest likelihood of being correct, and return it along
            with "
        "a confidence score and a verbal assessment explaining your judgment."
    )
)
```

Listing 7: Pseudo Code for Domain-Specific Multiple Choice QA

### C.3 PROMPT OPTIMIZATION

Prompt optimization is a critical component in leveraging large language models (LLMs) for complex tasks. The performance of LLMs is highly sensitive to variations in prompt structure, tone, and even the positioning of textual components. Minor changes—such as rephrasing imperative sentences or reordering blocks—can significantly impact the model's output.

Prompts typically follow a structured format, often divided into system and user sections. The system prompt provides general context, instructions, and expectations for the output, while the user prompt contains task-specific information. Common elements include task descriptions, constraints, few-shot examples, input/output format specifications, and guiding phrases like "Let's think step by step". These components are frequently organized using structured formats such as Markdown or JSON schemas.

**General Framework for Prompt Optimization** Ramnath et al. (2025) and Li et al. (2025) have summarized existing prompt optimization techniques into a generic prompt optimization framework. Algorithm 1 presented in Ramnath et al. (2025), formalizes the process of prompt optimization as follows. Given a task model $M_{\text{task}}$, an initial prompt $\rho \in V$, the goal of an prompt optimization system $M_{PO}$ is to obtain the best performing prompt-template $\rho^{opt}$ under a metric $f \in F$ and eval-set $D_{val}$ that maximizes expected performance:

$$\rho^{\text{opt}} := \arg\max_{\rho \in V} \mathbb{E}_{x \sim D_{\text{val}}} \left[ f \left( M_{\text{task}}(\rho \oplus x) \right) \right].$$

Since the objective function is not tractable due to the combinatorial nature of discrete token-sequence search spaces, the optimization process typically follows a generate-select-evaluate cycle, akin to local search algorithms (Russell & Norvig, 2020). Most methods begin with a predefined prompt template that specifies the structure and content to be included. An initial prompt may be constructed manually or generated automatically using an LLM.

Given a specific task, the dataset is usually partitioned into training and validation sets. The training set is used to optimize the prompt, while the validation set is employed for tuning hyperparameters. Additionally, a held-out test set is used to evaluate final performance. Candidate prompts are generated, filtered based on performance metrics, and refined over successive iterations, with incremental improvements guided by feedback from model outputs.

---

### Algorithm 1: Prompt Optimization Framework

1: $P_0 := \{\rho_1, \rho_2, \ldots, \rho_k\}$                  ▷ Initial seed prompts
2: $D_{val} := \{(x_1, y_1)\}_{i=1}^n$                ▷ Validation set
3: $f_1, \ldots, f_m \in F$               ▷ Inference evaluation
4: **for** $t = 1, 2, \ldots, N$ **do**            ▷ Iteration depth
5:      $G_t := M_{PO}(P, D_{val}, F)$       ▷ Generate prompt candidates with $M_{PO}$
6:      $P_t := Select(G_t, D_{val}, F)$       ▷ Filter and retain candidates
7:      **if** $f_{convergence} \leq \epsilon$ **then**     ▷ Optionally check for early convergence
8:          **exit**
9: **return** $\arg\max_{\rho \in P_N} E_{x \sim D_{val}} [f(M_{task}(\rho \oplus x))]$

---

**Parallelizing Prompt Optimization with Transduction Algebra**    Within the `Agentics` framework, the prompt function defined in Definition 9 maps type information into structured information objects for transduction. Prompt optimization in this context is naturally expressed using transduction algebra. Since candidate prompts can be generated by logical transduction, we adopt meta-prompt-based optimization strategies similar to those proposed in (Yang et al., 2024; Ye et al., 2024; Opsahl-Ong et al., 2024). Our approach emphasizes two key aspects:

- The `Agentics` framework supports a declarative style of prompting, where prompts are constructed to encode rich contextual and type-level information rather than procedural instructions.

- The optimization process follows the generate-select-evaluate cycle described in Algorithm 1. Importantly, the transduction algebra enables this optimization loop to be expressed in a functional and parallelizable manner. Prompt candidates can be generated and evaluated independently, allowing for efficient execution. This abstraction not only improves scalability but also decouples the optimization logic from the underlying execution strategy.

In summary, the `Agentics` framework provides a principled and extensible foundation for prompt optimization. It integrates declarative prompt construction, transduction algebra, and parallel search strategies into a unified system that supports both expressiveness and scalability. Next, we present a functional design pattern for implementing prompt optimization using transduction algebra, abstracting away procedural details common in existing approaches. In Section A.6, our experiments demonstrate that declarative context optimization improves performance and that parallelization yields substantial runtime gains.

**Data Models**    We begin by defining two data models using Pydantic: `OptimizationTask` and `GSM8K`. The `OptimizationTask` schema captures the components of a prompt template—such as role, goal, expected output, and imperative instructions—along with a score field for evaluation. The `GSM8K` schema represents the target task, including the question, ground-truth answer, model-generated reasoning, and correctness flag.

```python
class OptimizationTask(BaseModel):
    # a list of demo tasks used to guide prompt generation
    demos: list[Any] = Field(description="optimization demo tasks")
    # role description to be embedded in the prompt (e.g., 'You are a math tutor')
    role: str = Field(description="New role instruction")
    # goal statement describing what the prompt aims to achieve
    goal: str = Field(description="New goal instruction")
```

```
 8    # criteria for what constitutes a good or acceptable output
 9    expected_output: str = Field(description="New expected_output instruction")
10    # imperative phrase (e.g., 'Let's think step by step')
11    imperative: str = Field(description="New imperative")
12    # evaluation score assigned to the prompt after testing on validation data
13    score: int = Field(description="evaluation score")
14
15 class GSM8K(BaseModel):
16    question: str = Field(description="a grade school math question.")
17    answer: str = Field(description="the ground-truth answer")
18    response_think: str = Field(description="the step by step reasoning")
19    response_answer: str = Field(description="the final answer")
20    # boolean flag indicating whether the response answer is correct
21    correct: bool
```

Listing 8: Data Models for GSM8K Prompt Optimization

**Meta Prompt and Optimized Prompts** The meta-prompt (OPT_META_INSTRUCTION) guides the generation of new prompt templates by describing the structure and expectations for the optimizer. It includes historical context from previous iterations and instructs the model to avoid redundancy. The user prompt template (USER_PROMPT_TEMPLATE) is instantiated with the optimized parameters and used to evaluate candidate prompts on the validation set.

The following shows the meta-prompt used for optimizing the prompt template for the GSM8K dataset.

```
 1  OPT_META_INSTRUCTION = "Your proposed prompt template will be used in the following way.
 2  * You are "role" -- this role must be suitable for solving the demo task.
 3  * Your personal goal is: "goal" -- the goal achieves the outputs given inputs.
 4  * This is the expected criteria for your final answer "expected_output" -- this constrains the
       output format.
 5  * You can add a short imperative instruction "imperative" -- this comes after the input of the
       task.
 6
 7  [[Several demo tasks of input and outputs will be provided when you solve problem.]]
 8
 9  [[The previous optimized prompt templates with scores appear from the worst to the best.]]
10  {optimization_history}
11
12  * Given the previous optimization results, don't generate duplicate or similar prompt
       templates.
13  * Generate prompt template that achieves the best score, and succint and concise instructions.
14  "
15
16
17  USER_PROMPT_TEMPLATE = "
18  You are {role}.
19  Your personal goal is: {goal}.
20  This is the expected criteria for your final answer: {expected_output}.
21
22  solve the following task.
23  {question}
24
25  {imperative}
26  "
```

Listing 9: Meta Prompt and Template for GSM8K Prompt Optimization

In the following, we show the prompts returned by APO, including both system and user prompts. The system prompt consists of three components: role, goal, and expected_output. In the user prompt, an imperative statement appears after each question. A score of 89 was evaluated on the validation set, using 100 problems sampled from the training set, which is higher than the test score of 85.

```
1  {
2  "role": "Elite Mathematical Problem Solver",
3  "goal": "To swiftly and accurately determine the precise numerical solution by meticulously
       dissecting complex problem statements, identifying pivotal information, and applying a
       comprehensive array of advanced mathematical concepts, formulas, and logical reasoning to
        achieve an optimal solution, ensuring efficiency, precision, and reliability in all
       calculations, while providing accurate and relevant answers",
4  "expected_output": "A single, exact numeric value that directly addresses and solves the given
        mathematical problem, ensuring all calculations are correctly executed, based on the
       information provided, and presented in a clear, concise manner, with strict adherence to
       mathematical rules and consideration of all given data",
5  "imperative": "Thoroughly analyze the problem statement, extract key information, apply
       pertinent mathematical principles and logical reasoning to derive the accurate numerical
       answer, and provide the final answer in the required numeric format, while validating the
        solution through re-evaluation of calculations and verification of the accuracy of the
       obtained result, and ensuring precision, accuracy, and reliability in all steps of the
       calculation process.",
6  "score": 89
```

```
7   }
```

Listing 10: Optimized Prompt for GSM8K using Llama-3.3-70B Model

The following result is obtained from the Qwen3-8B model. A score of 98 was evaluated on the validation set, using 100 problems sampled from the training set. This score is higher than the test score of 92.

```
1   {
2   "role": "Math Problem Solver",
3   "goal": "Solve complex problems by decomposing them into sequential steps, applying arithmetic
        operations (percentages, fractions, averages), ensuring unit consistency, and presenting
        the final answer in a boxed format.",
4   "expected_output": "A single numeric value boxed (e.g., \\boxed{64}) representing the solution
        , derived through precise step-by-step calculations with attention to percentages,
        fractions, and averages.",
5   "imperative": "Analyze the problem statement, execute calculations step-by-step with focus on
        percentages, fractions, and unit conversions, then box the final numeric result.",
6   "score": 98
7   }
```

Listing 11: Optimized Prompt for GSM8K using Qwen3-8B Model

The following shows the meta prompt for optimizing the prompt template for MCQA dataset.

```
1   OPT_META_INSTRUCTION = "Your proposed prompt template will be used in the following way.
2   * You are "role" -- this role must be suitable for solving the demo task.
3   * Your personal goal is: "goal" -- the goal achieves the outputs given inputs.
4   * This is the expected criteria for your final answer "expected_output" -- this constrains the
        output format.
5   * Extract"task_context" from demo tasks to explain the problem context -- this comes before
        the input of the task.
6   * You can add a short imperative instruction "imperative" -- this comes after the input of the
        task.
7
8
9   [[Several demo tasks of input and outputs will be provided when you solve problem.]]
10
11  [[The previous optimized prompt templates with scores appear from the worst to the best.]]
12  {optimization_history}
13
14  * Given the previous optimization results, don't generate duplicate or similar prompt
        templates.
15  * Generate prompt template that achieves the best score, and succint and concise instructions.
16  "
17
18  # Template used to instantiate a user prompt from the optimized parameters.
19  # This is applied to each validation example.
20  USER_PROMPT_TEMPLATE = "
21  You are {role}.
22  Your personal goal is: {goal}.
23  This is the expected criteria for your final answer: {expected_output}.
24
25  This is the general task context.
26  {task_context}
27
28  solve the following task.
29  {question}
30  {options}
31  {option_ids}
32  {asset_name}
33  {relevancy}
34  {question_type}
35  {subject}
36
37  {imperative}
38  "
```

Listing 12: Meta Prompt and Template for MCQA Prompt Optimization

In the following, we show the system and user prompts returned by APO. The system prompt includes three components: role, goal, and expected_output. In the user prompt, task_context appears before each question, and an imperative statement follows each question. The test score was 54.

```
1   {
2   "role": "Industrial Asset Diagnostician",
3   "goal": "To accurately identify the most relevant sensors for detecting specific failure modes
        in various industrial assets and determine the least relevant failure events for
        abnormal sensor readings, optimizing asset performance and reducing downtime",
4   "expected_output": "A concise description of the most relevant sensor for monitoring a
        specific failure mode or the least relevant failure event that does not significantly
        contribute to detecting a particular failure mode in the given asset, including the
        option id or description",
```

```
 5  "task_context": "The task involves analyzing the relationship between sensor readings, failure
        modes, and industrial assets such as steam turbines, aero gas turbines, electric
        generators, and compressors, to determine the most relevant sensors for monitoring
        specific failure modes and the least relevant failure events for abnormal sensor readings
        ", "imperative": "Analyze the provided asset, sensor readings, and failure modes to
        identify the most relevant sensor for monitoring a specific failure mode or the least
        relevant failure event for an abnormal sensor reading, considering the relationship
        between failure modes, sensors, and assets",
 6  "score": 54}
```

Listing 13: Optimized Prompt for MCQA using Llama-3.3-70B Model

**Main Algorithm** The main optimization loop follows the generate-select-evaluate cycle described in Algorithm 1. It begins by preparing the training and validation sets using the `Agentics` (`AG[GSM8K]`). Demo tasks are extracted and transformed into `OptimizationTask` instances.

In each iteration, candidate prompt templates are generated via self-transduction using the meta-prompt. These templates are applied to the validation set using the user prompt format. The responses are evaluated using the grading function defined in `GSM8K.grade`, and scores are assigned. The best-performing prompts are retained using a filtering function (`keep_best_k`). This loop continues until convergence or a maximum number of iterations is reached. The use of transduction and asynchronous execution enables parallel evaluation of prompt candidates, improving scalability and runtime efficiency.

```
 1  # load GSM8K training data into \texttt{Agentics} abstraction
 2  trainset = AG.from_json("gsm8k_train.jsonl")
 3  # truncate to a subset for training
 4  trainset = trainset.truncate_states(num_trains)
 5  # create demo tasks from training examples
 6  demosets = create_optimization_demos(trainset, num_demos)
 7  # convert demo tasks into OptimizationTask instances
 8  optimization_tasks = OptimizationTask.create_optimization_tasks(demosets)
 9  # prepare validation set from remaining examples
10  validationset = trainset.truncate_states(num_trains, num_trains + num_devs)
11
12  # initialize optimizer AG[OptimizationTask] with demo tasks
13  optimizer = AG.from_states(optimization_tasks, atype=OptimizationTask)
14
15  # set default parameters and prompt configuration
16  set_default_params(optimizer)
17  optimizer.prompt_template = "{{"demo tasks":{demos}}}"
18  optimizer.prompt_params = {"role": "Prompt optimizer.", "goal": "Propose diverse prompt
        templates that achieves high performance for the demo task given as input.", "backstory":
         "Understand the problem domain given the demo task example and propose what answer
        should be generated.", "expected_output": "the outputs are role, goal, and the expected
        output description, and imperative sentence for solving provided tasks."}
19
20  # initialize list to store optimized prompt tasks
21  optimized_tasks = []
22  for iter_ind in range(max_iter):
23      # generate candidate prompt templates using meta-prompt and transduction
24      # transduction from demos to prompt parameters
25      optimizer.instructions = OPT_META_INSTRUCTION.format(
26          optimization_history = get_history_string(optimized_tasks))
27      optimizer = asyncio.run(optimizer.self_transduction(
28          ["demos"],
29          ["role", "goal", "expected_output", "imperative"]))
30
31      # apply candidate prompts to validation set using user prompt format
32      opt_eval = optimizer * validationset
33      opt_eval.prompt_template = USER_PROMPT_TEMPLATE
34      opt_eval = asyncio.run(opt_eval.self_transduction(
35          ["role", "goal", "expected_output", "imperative", "question"],
36          ["response_think", "response_answer"]))
37
38      # evaluate responses using GSM8K grading function
39      executed_tasks = opt_eval / validationset
40      for ind, exectask in enumerate(executed_tasks):
41          exectask = asyncio.run(exectask.amap(GSM8K.grade))
42          setattr(optimizer[ind], "score", summary["score"])
43
44      # retain top-performing prompts for next iteration
45      optimized_tasks.extend(optimizer.states)
46      optimized_tasks = keep_best_k(optimized_tasks)
```

Listing 14: Pseudo Code for GSM8K Prompt Optimization

