# OpenReview forum: "Transduction is All You Need for Structured Data Workflows"
_ICLR.cc/2026/Conference — ICLR 2026 Conference Desk Rejected Submission_

### Official Review · Reviewer_jijG · 2025-10-27

**Soundness:** 3
**Presentation:** 4
**Contribution:** 3
**Rating:** 6
**Confidence:** 4

**Summary:**

This paper introduces Agentics, a novel functional AI framework designed to address the brittleness and lack of formal semantics in current LLM-based systems for structured data workflows. The core concept is logical transduction, a data-centric paradigm where agents act as stateless transducers that transform data between strongly-typed schemas. This approach shifts the focus from conversational, prompt-chained interactions to a more principled, type-driven computation.
The framework is formalized through a Logical Transduction Algebra" (LTA), which guarantees key properties like conditional determinism, statelessness, and compositionality. It is implemented with an asynchronous map-reduce style programming model, enhancing modularity, parallelism, and reproducibility. Through experiments on tasks like text-to-SQL, schema matching, and domain-specific question answering, the authors demonstrate that Agentics achieves competitive or improved performance. Notably, it shows significant gains in robustness against knowledge-invariant perturbations, underscoring its reliability. The paper advocates for a paradigm shift towards a more formal, data-centric approach for building reliable and scalable data workflows with LLMs.

**Strengths:**

Rather than offering an incremental improvement on existing frameworks, the paper introduces a genuine paradigm shift, moving from chat-centric models to a principled, data-centric functional approach. The introduction of a formal Logical Transduction Algebra is a profound and novel contribution, as it grounds the behavior of LLM agents in a mathematically rigorous foundation—a rarity in this empirically-driven
field. This theoretical rigor directly translates into the work's high quality. The LTA is not merely conceptual; it is formally defined with proven properties, which provides a sound basis for the claims. The empirical validation is equally strong, moving beyond standard accuracy metrics. The inclusion of robustness tests against knowledge-invariant perturbations is particularly compelling, as it provides direct, powerful evidence for the central hypothesis that this approach mitigates the "brittleness" of LLM pipelines.

**Weaknesses:**

1. Lack of Direct Comparative Evaluation with Existing Frameworks: The most significant weakness is the absence of a direct, quantitative benchmark against the very frameworks it critiques, such as LangChain. The paper makes strong claims about overcoming their brittleness and lack of formal semantics. While the internal experiments and ablation studies are valuable, they only compare variations of the Agentics approach against a baseline. To truly substantiate the superiority of this new paradigm, an end-to-end implementation of a complex, multi-step workflow in both Agentics and a state-of-the-art alternative framework would be necessary. Such a comparison could measure key metrics like final task accuracy, error recovery rate, execution speed, and perhaps even lines of code or development complexity, providing much stronger evidence for its claims.
2. Unaddressed Limitations of the Stateless Paradigm: The framework's core strength—statelessness—is also its primary limitation. This design choice is fundamental to achieving parallelism and reproducibility. However, many real-world data workflows are inherently stateful. For example, a workflow might need to aggregate information over a series of steps where the context from step 1 is needed in step 5, or it might need to maintain a conversational memory in an interactive data-cleaning session. The paper does not adequately discuss the boundaries of its paradigm or offer clear patterns for handling state when it is unavoidable. This leaves a critical gap in understanding the practical applicability of Agentics for a whole class of important problems.

**Questions:**

1. On Comparative Evaluation: Could you elaborate on the decision not to include a direct benchmark against a mainstream agentic framework like LangGraph? We believe that even a single, detailed case study comparing the implementation of one of the paper's complex workflows in Agentics versus a state-of-the-art alternative would powerfully highlight the practical benefits in terms of robustness, code clarity, and performance.
2. On Handling State: The stateless nature of transduction is a cornerstone of the framework, enabling parallelism and reproducibility. However, how does the Agentics paradigm envision handling workflows that intrinsically require state to be maintained and passed between non-adjacent steps? Are such workflows considered out-of-scope, or are there recommended design patterns to manage this?

---

> ### Author Response · Authors · 2025-11-21
> **Lack of direct comparative evaluation with existing framework.**
>
> We sincerely thank the reviewer for their insightful feedback.
> We appreciate your recognition of the proposed software framework as a novel functional AI framework, and precise assessment of the scope of the work as well as contributions.
> Below, we address each of your comments and questions with detailed responses.
>
> ### Comparison with Other Frameworks
> Thanks very much for your question and pointing out possible case studies that we can elaborate.
> We believe this question is very important and related to all other reviewers' comment, we decided to move the answer to the overall comment.
>
> In the global response, we provide the comparison with DSPy and Langgraph in text-to-sql use case. We discuss the motivation and advantages with the concrete code examples from three frameworks. Through the comparison, we see that the Agentics approach has advantage in composition of dataworkflows in a finer-grained than the other two and it results in better scaling up in terms of the throughput, running time, and error handling due to LLM api call errors, which frequently happens in practice.
>
>
> ### Could you elaborate on the decision not to include a direct benchmark against a mainstream agentic framework like LangGraph?
> The main scope of this paper is a software framework for LLM applications, commonly referred to as agentic AI.
> We present
> (1) the proposed programming model,
> (2) a summarized version of Logical Transduction Algebra (with full details in the appendix), and
> (3) use cases in the experiment section.
>
> The experiments demonstrate that the proposed programming model applies to diverse structured data workflow tasks. We include simplified Python code examples and report performance primarily in terms of accuracy, acknowledging that accuracy alone is not a perfect fit for evaluating programming models.
> To address scalability, Section 4.2 and Appendix A.5 (MCQA) and A.6 (Prompt Optimization) include running-time experiments showing improvements with batch sizes up to 10. This result depends on the computing environments such as settings of backend LLM servers or GPU configurations.
>
> From our experience implementing these applications in other frameworks, we observed that compositionality and scaling performance improve significantly in Agentics. For example, processing over 1,000 text-to-SQL problems completes in about one hour with Agentics, whereas DSPy implementations take several hours.
> However, we note that systematic runtime benchmarking is challenging due to numerous factors influencing performance. If runtime is ignored, two implementations can achieve similar accuracy by generating identical prompts for LLMs.
>
> To address reviewer feedback, we have included code examples from three frameworks (Agentics, DSPy, LangGraph) for implementing text-to-SQL, highlighting differences in composability, and execution models. Please let us know if further clarification or additional comparisons are needed.

---

> ### Author Response · Authors · 2025-11-21
> **On Limitation of stateless approach**
>
> Our design explicitly targets structured workflows that are schema-constrained, and composable. In part, the proposed framework is inspired by the functional programming style, which doesn't allow state changes but creates a new data instance after applying a function. The logical transduction is stateless, and LLM execution is as well.
>
> As the reviewer mentioned, many real-world data workflows are stateful. The proposed framework can handle such cases by designing the transduction pipeline with functions and tools, including external stores.
>
> For a concrete example, let's consider a transduction pipeline that transforms data objects of type $T_1$.
> The agentic structure $AG[T_1]$ holds states of type $T_1$.
> We transform the objects of type $T_1$ to $T_2$ via logical transduction,
> yielding $AG[T_2]$ with states of type $T_2$.
> For the next transduction to obtain $AG[T_3]$, if we need the context from $AG[T_1]$ which we project to $AG[Z]$
> or the transformation of the context via MapReduce from $AG[T_1]$ yielding $AG[Z]$,
> we can store $AG[Z]$ in memory and access it before transduction to obtain $AG[T_3]$.
> To achieve this, the context from $AG[Z]$ can be brought into $T_2$,
> by mutating the data type from $T_2$ to $T_2'$ by using the type operators,
> which essentially extending the data types with $Z$.
> This could be as simple as adding a field from $Z$ to $T_2$ to get $T_2'$.
>
> Due to the space limitation, we briefly mentioned type manipulations in the main paper. Appendix B provides more details. For type manipulation, we provide Definition 2, which applies set operations to the fields of the data schema.
>
> Note that the algebra in the Appendix applies to the Agentic structure, essentially a list of typed objects, and the Monoid is shown to the Agentic instances with a binary operator that concatenates two list states of the same type. The product and quotient of the Agentic structure is also defined appropriately for the list of states with two types. This algebra justifies manipulating typed objects or states, and the logical transduction operator fills in the values to the fields in the target type.
>
> As the reviewer mentioned, the basic programming patterns in Agentics aim to exploit parallelism. An exercise on the prompt optimization in Appendix C.3 shows how local search for prompt optimization can be implemented by using the product and quotient on the Agentic structure.

---

### Official Review · Reviewer_PX9e · 2025-10-30

**Soundness:** 2
**Presentation:** 2
**Contribution:** 3
**Rating:** 8
**Confidence:** 1

**Summary:**

This paper propose a framework for processing LLM-based structured data workflow pipelines. The authors evaluate Agentics on tasks such as schema matching, text-to-SQL parsing, data imputation, and domain-specific multiple-choice question answering (MCQA), showing competitive or improved performance compared to baselines.

**Strengths:**

- The introduction of Logical Transduction Algebra (LTA) provides a principled, algebraic framework for composing LLM-based workflows. This formalism is a significant contribution, offering reproducibility, statelessness, and composability.
- The asynchronous MapReduce-style programming model (aMap/aReduce) enables efficient parallel execution, which is crucial for large-scale data workflows.

**Weaknesses:**

- The framework is highly optimized for structured data workflows but may not generalize well to open-ended, context-heavy, or creative tasks that require rich contextual understanding or instruction-following.
- For some tasks (e.g., schema matching), the baselines are limited to GPT-3.5. Comparing against more recent or task-specific models could strengthen the claims.

**Questions:**

The reviewer is not familiar with the logical transduction area and therefore CANNOT make a fair judgement of the technical merits of the paper. Please refer to other reviewers for the detailed concerns.

---

> ### Author Response · Authors · 2025-11-21
> **On weakness of the framework (highly optimized for structured data workflows)**
>
> We sincerely thank the reviewer for generous ratings and requests on improving the experiment section.
> We appreciate your recognition of the algebraic framework for composing LLM-based workflows.
>
> ### strength and weakness of the framework
> We agree to the reviewer's perspective. In the overall comment, we provide the comparison with DSPy and Langgraph in text-to-sql use case. We show the design goals and advantages with the concrete code examples from three frameworks.
>
> One could argue that Agentics could be used for implementing creative tasks with dynamic human-interactive experience, but we clearly see that it is not aligned with the design goal of the framework and it loses the main advantages. This is similar to other frameworks. Therefore, we better to rephrase that there is no single software framework that dominates others in all requirements.
> Through the comparison, we see that the Agentics approach has advantage in composition of data workflows in a finer-grained level than the other two and it results in better scaling up in terms of the throughput, running time, and error handling due to LLM api call errors, which frequently happens in practice.
>
>
> ### Additional experiment with DiscoveryBench
> we also added an experiment with more recent DiscoveryBench in the global response, showing that agentics framework offers the state-of-the art performance on one of the task with GPTOSS120b compared with existing approaches using GPT-4o.

---

> > ### Comment · Reviewer_PX9e · 2025-11-26
> >
> > Thank you for the explanations. I encourage the authors to make the paper easier to understand for audience outside this domain, which is also mentioned by other reviewers that the low-level details about the mathematic formulations are overwhelming.

---

> ### Author Response · Authors · 2025-12-03
> **On baseline results from GPT-3.5**
>
> The data workflow tasks in the paper, schema matching, text-to-SQL, and data imputation are related to data management applications, and we take representative papers from the literature. This application domain is close to enterprise use cases, but there are no established benchmarks proposed in the literature yet.
>
> The baseline prompts for the schema matching and data imputation tasks were evaluated with the GPT-3.5 model when the paper was published. We believe this issue can be addressed by both clarifying the role of our initial baselines and by providing additional experimental comparison in a fair setting.
>
> ## Schema matching Task
> To provide a fair comparison, we evaluated the exact same settings in the baseline paper (Parciak et al., 2024b) using GPTOSS-120B.
> The table below shows the results of the schema matching task (F1 score).
>
> | Datasets | **1-to-1** | | | **1-to-N** | | |
> | :--- | :--- | :--- | :--- | :--- | :--- | :--- |
> | | **GPT 3.5 Baseline** | **GPTOSS-120B Baseline** | **Agentics** (gpt-oss & llama4) | **GPT 3.5 Baseline** | **GPTOSS-120B Baseline** | **Agentics** (gpt-oss & llama4) |
> | :--- | :--- | :--- | :--- | :--- | :--- | :--- |
> | AdCO | 0.000 | 0.000 | 0.500 | 0.133 | 0.000 | 0.250 |
> | AdVD | 0.000 | 0.000 | 0.330 | 0.083 | 0.333 | 0.250 |
> | AdVO | 0.235 | 0.000 | 0.400 | 0.320 | 0.286 | 0.143 |
> | DiCO | 0.667 | 0.667 | 0.500 | 0.800 | 0.500 | 0.667 |
> | LaMe | 0.471 | 0.200 | 0.364 | 0.500 | 0.364 | 0.500 |
> | PaPe | 0.571 | 0.000 | 0.000 | 0.500 | 0.400 | 0.667 |
> | PrDE | 0.222 | 0.333 | 0.000 | 0.417 | 0.444 | 0.211 |
> | SeVD | 0.000 | 0.000 | 0.500 | 0.400 | 0.000 | 0.400 |
> | TrVD | 0.000 | 0.000 | 0.333 | 0.429 | 0.182 | 0.600 |
> | **Mean** | 0.241 | 0.133 | 0.325 | 0.398 | 0.279 | 0.382 |
>
>
> We can see that the baseline performance has dropped from 0.241 to 0.133 for the 1-to-1 case, and from 0.398 to 0.279 in the 1-to-N case.
>
> The baseline approach can be considered as a pure prompt-based implementation of schema matching, and the Agentics approach leverages type-driven, structured prompting implemented by a functional style program. In the appendix, we also provided the code for the Agentics implementation, and the baseline implementation is available from the original author’s paper.
>
> In summary, the schema matching experiment with re-evaluation on GPTOSS-120B model shows the benefit of the programming model based on the logical transduction algebra, and improved performance due to Pydantic types and structured prompts.
>
> We would like to emphasize that the main contribution of this paper is not on surpassing the performance on existing benchmarks, but on proposing a new program model suitable for structured data workflow tasks.
>
> ## Data imputation task
> In case of the data imputation task, we requested the original authors to share the original prompts used in the paper, but we couldn’t hear back. In the Agentics implementation, the data imputation task is framed as a problem of finding proper missing values given the full examples in the in-context examples, which is likely to be a much more challenging setting than the original pape,r that may limit the possible choice of labels.
>
> Due to the non-reproducibility of the baselines in a modern setting, we added a new modern benchmark, DiscoveryBench.  DiscoveryBench requires scientific semantic reasoning to decide on appropriate analysis techniques for a specific domain (e.g., spatial autocorrelation for plant invasion) and handle steps like data cleaning, normalization, and mapping goal terms to dataset terms. Task solutions typically require a multi-step workflow. We show the detailed comparison in the global response summarizing the performance from the baselines as well as the Agentics-based program.

---

> ### Author Response · Authors · 2025-12-03
> **On open-ended, context-heavy, or creative tasks**
>
> The proposed framework was primarily targeted at solving structured data workflow tasks. We also show that the proposed approach is useful in an open-ended task, DiscoveryBench. DiscoveryBench requires scientific semantic reasoning to decide on appropriate analysis techniques for a specific domain (e.g., spatial autocorrelation for plant invasion) and handle steps like data cleaning, normalization, and mapping goal terms to dataset terms. Task solutions typically require a multi-step workflow.
>
> We show the detailed comparison in the global response, summarizing the performance from the baselines as well as the Agentics-based program. This experiment implies that the proposed framework is also useful for handling tasks beyond simple data transformations.

---

### Official Review · Reviewer_Bc8M · 2025-10-30

**Soundness:** 3
**Presentation:** 2
**Contribution:** 2
**Rating:** 4
**Confidence:** 4

**Summary:**

The authors of this paper propose a functional agentic AI framework Agentics, which is designed for building LLM-based structured data workflow pipelines. Its core contribution is a shift from conversation-centric to a data-centric paradigm, where agents are modeled as stateless transducers operating over well-defined, Pydantic-based types. This paper formalizes this approach with a Logical Transduction Algebra (LTA), which provides a compositional calculus for building, analyzing, and optimizing these pipelines. The proposed framework is implemented as a Python library and evaluated on a range of structured data tasks, including schema matching, text-to-SQL, data imputation, and domain-specific multiple-choice QA, demonstrating competitive or improved performance due to its structured prompting and asynchronous execution model.

**Strengths:**

1. This work is significant for addressing the critical challenge of building robust, composable, and scalable data workflows with LLMs, offering a principled alternative to fragile, chat-based agent systems.

2. The proposed "logical transduction" paradigm is well executed, which elegantly re-frames LLM agents as stateless transducers over typed data. It combines a sound theoretical foundation (provides formal definitions with proven properties) with a practical Python implementation, demonstrating strong empirical results across diverse structured data tasks.

3. The framework is evaluated across multiple domains and tasks, including schema matching, text-to-SQL (BIRD, Archer), data imputation, and domain-specific multiple-choice QA, demonstrating its broad applicability.

**Weaknesses:**

1. This paper's motivation is not convinced and clear. It never truly clarifies why existing agent frameworks are fundamentally ill-suited for structured data, offering no tangible examples of where they fail and why a new paradigm is needed.

2. The experiments are not sufficient to verify the proposed framework's advantage, since the baselines are really weak (GPT-3.5), missing the comparisons with recent frameworks like DSPy or AutoGen on the same tasks.

3. The paper frames LTA as a major contribution, but the algebra is relatively straightforward, essentially applying monoid operations to typed data structures with LLM inference. The claimed properties are mainly inherited from the underlying components rather than emergent from the algebra itself.

4. The presentation is somewhat difficult to follow, with excessive formalism in early sections that could be streamlined. The connection between theoretical LTA and practical implementation is unclear. Code examples mix pseudo-code with Python syntax inconsistently.

**Questions:**

Miner issues:
1. Line 486: "Ethics Statement" should be in a separate \section.

2. The composite workflow (P+FS+KW+SQ+SL+OP) achieves 10.33% improvement, but individual components show negative or minimal gains. Can you provide more analysis on why the combination works better?

---

> ### Author Response · Authors · 2025-11-21
> **Motivation, why existing agent frameworks are not well-suited for the structured data workflows.**
>
> We sincerely thank the reviewer for their insightful feedback.
> We appreciate your recognition of the proposed software framework is significant for addressing the critical chanlleges in scalable data workflows with LLMs.
> Below, we address each of your comments and questions with detailed responses.
>
>
> Existing agentic AI frameworks adopt the chat-centric paradigm that results in an event-driven architecture in which agents or users exchange natural language-based messages. This is a natural choice for the open-ended interactive tasks.
>
> The structured data workflow tasks typically comprised of common tasks for data collection, extraction, cleansing, transformation, and analysis. They adhere on strict data types or schema and leave less room for human-in-the-loop experience due to the large scale batch processing. For example, date-format correction, applying suitable transformation to data columns, etc.
>
> Such data processing pipelines are relatively stable, unlike chat-based conversation that can dynamically vary depending on the conversation. In addition, composition of multiple transformations are typical. The proposed approach addresses this aspects by treating typed objects (pydantic models) as the primary objects and the framework is designed to support transformation of the typed objects with large language model, which is called logical transduction.
>
> As a concrete example, we could consider a data workflow task that applies data transformation tasks that extract a subset of data given a schema and apply schema mapping from the meta-data to a desired set of concepts or ontology (A simplified version of the schema machting usecase is presented in the experiment section.)
> The main properties of this task are
> (1) all data in this workflow strictly follows the schema constraints,
> (2) each processing of individual workload is independent to each other and does not depend on earlier transformations (stateless), and
> (3) different transformations can be composed for example, mapping the meta-data to different concept/terminology and chaning format of the data or translate one language into other language.
>
> The chat-based approach can also implement such pipelines but we see that it is not suitable because
> (1) schema-constrained transformation can be better handled in a programmable way, and
> (2) the large volume of processing indepdent relatively small tasks is cumbersome through chats.
> When one of the transformation failed (which happens often when we use LLMs), we only need to retry on the failed transduction that is independent to other workloads.
>
> In the overall comment, we provide the comparison with DSPy and Langgraph in text-to-sql use case.
> We discuss the motivation and advantages with the concrete code examples from three frameworks.
>
> Please check out the comparison part of the rebuttal.
> Through the comparison, we see that the Agentics approach has advantage in composition of dataworkflows in a finer-grained than the other two and it results in better scaling up in terms of the throughput, running time, and error handling due to LLM api call errors, which frequently happens in practice.

---

> ### Author Response · Authors · 2025-11-21
> **On logical transduction algebra and its contribution.**
>
> As reviewer mentioned, we also see that the algebra itself is straightforward, and we don’t claim new mathematical findings or the noveltiy in the type operators or monoidal constructions. The main contribution of this paper is on software libraries for agentic AI, and the logical transduction algebra provides a formal basis for programming models.
> In the appendix, we define the structure over typed objects, called states, the operations over the types, the connection to the map/reduce programming models, the transformation of the states from one type to another by LLM, called logical transduction.
> Those formal elements are implemented in the framework and we show programming model examples illustrating how they are used.

---

> ### Author Response · Authors · 2025-12-03
> **On baseline results from GPT-3.5**
>
> The data workflow tasks in the paper, schema matching, text-to-SQL, and data imputation are related to data management applications, and we take representative papers from the literature. This application domain is close to enterprise use cases, but there are no established benchmarks proposed in the literature yet.
>
> The baseline prompts for the schema matching and data imputation tasks were evaluated with the GPT-3.5 model when the paper was published. We believe this issue can be addressed by both clarifying the role of our initial baselines and by providing additional experimental comparison in a fair setting.
>
> ## Schema matching Task
> To provide a fair comparison, we evaluated the exact same settings in the baseline paper (Parciak et al., 2024b) using GPTOSS-120B.
> The table below shows the results of the schema matching task (F1 score).
>
> | Datasets | **1-to-1** | | | **1-to-N** | | |
> | :--- | :--- | :--- | :--- | :--- | :--- | :--- |
> | | **GPT 3.5 Baseline** | **GPTOSS-120B Baseline** | **Agentics** (gpt-oss & llama4) | **GPT 3.5 Baseline** | **GPTOSS-120B Baseline** | **Agentics** (gpt-oss & llama4) |
> | :--- | :--- | :--- | :--- | :--- | :--- | :--- |
> | AdCO | 0.000 | 0.000 | 0.500 | 0.133 | 0.000 | 0.250 |
> | AdVD | 0.000 | 0.000 | 0.330 | 0.083 | 0.333 | 0.250 |
> | AdVO | 0.235 | 0.000 | 0.400 | 0.320 | 0.286 | 0.143 |
> | DiCO | 0.667 | 0.667 | 0.500 | 0.800 | 0.500 | 0.667 |
> | LaMe | 0.471 | 0.200 | 0.364 | 0.500 | 0.364 | 0.500 |
> | PaPe | 0.571 | 0.000 | 0.000 | 0.500 | 0.400 | 0.667 |
> | PrDE | 0.222 | 0.333 | 0.000 | 0.417 | 0.444 | 0.211 |
> | SeVD | 0.000 | 0.000 | 0.500 | 0.400 | 0.000 | 0.400 |
> | TrVD | 0.000 | 0.000 | 0.333 | 0.429 | 0.182 | 0.600 |
> | **Mean** | 0.241 | 0.133 | 0.325 | 0.398 | 0.279 | 0.382 |
>
>
> We can see that the baseline performance has dropped from 0.241 to 0.133 for the 1-to-1 case, and from 0.398 to 0.279 in the 1-to-N case.
>
> The baseline approach can be considered as a pure prompt-based implementation of schema matching, and the Agentics approach leverages type-driven, structured prompting implemented by a functional style program. In the appendix, we also provided the code for the Agentics implementation, and the baseline implementation is available from the original author’s paper.
>
> In summary, the schema matching experiment with re-evaluation on GPTOSS-120B model shows the benefit of the programming model based on the logical transduction algebra, and improved performance due to Pydantic types and structured prompts.
>
> We would like to emphasize that the main contribution of this paper is not on surpassing the performance on existing benchmarks, but on proposing a new program model suitable for structured data workflow tasks.
>
> ## Data imputation task
> In case of the data imputation task, we requested the original authors to share the original prompts used in the paper, but we couldn’t hear back. In the Agentics implementation, the data imputation task is framed as a problem of finding proper missing values given the full examples in the in-context examples, which is likely to be a much more challenging setting than the original pape,r that may limit the possible choice of labels.
>
> Due to the non-reproducibility of the baselines in a modern setting, we added a new modern benchmark, DiscoveryBench.  DiscoveryBench requires scientific semantic reasoning to decide on appropriate analysis techniques for a specific domain (e.g., spatial autocorrelation for plant invasion) and handle steps like data cleaning, normalization, and mapping goal terms to dataset terms. Task solutions typically require a multi-step workflow. We show the detailed comparison in the global response summarizing the performance from the baselines as well as the Agentics-based program.

---

> ### Author Response · Authors · 2025-12-03
> **On Text-to-SQL Experiment**
>
> We analyzed how the combination of the components results in the final performance in more detail, and we find that the combination of the two, Schema Linking (SL) + Optimized Prompts (OP) is contributing the most from the naive prompting configuration (P).
>
> Below is the summary of the results based on the LLama3-70B and LLama4-17B models.
> * llama 3: Mean 0.59662, Std 0.002121
> * llama 4: Mean 0.59324,  Std 0.05059
>
> In addition, we expanded the evaluation with frontier models, which should be able to leverage the in-context data better. **Claude Sonnet 4.5** was tested on the schema linking + optimized prompts, and compared to including all the additional in-context data.
> * Schema Linker (SL) + Optimized Prompt( OP): Mean 0.6653, Std 0.005183
> * All components: Mean 0.6653, Std 0.005183
>
> Given those two components, other random few-shot examples, keywords, and subquestions are not contributing as much. This is also consistent in the research in text-to-SQL that the in-context example should be given carefully to achieve better performance.
>
> Note that over 66% of the Bird-dev set execution match performance comes from a single generation, without more complicated techniques proposed in recent literature, such as test-time scaling, reflection/refinement, chain-of-thought prompting, dynamic few-shots, or consistency decoding.

---

### Author Response · Authors · 2025-11-21
**Global response to all reviewers**

We thank all reviewers for their efforts in reviewing our submission, and insightful comments and constructive feedback.
We will address the questions and comments individually to reviewers, but we believe the comparison of programs implemented in different frameworks is worth being shown in the global response.
The comparison was suggested by reviewer jijG, and it will also address other issues requesting clear motivation by Bc8M, weak suitability for creativity tasks by reviewer PX9e.


Reflecting the comments on the experiments, we would like to add an additional experiment with DiscoveryBench (Bodhisattwa Prasad Majumder, et al, 2024) to address the concerns on the weakness of the experiment raised by reviewers (PX9e, Bc8M).
In addition, we also improve the experiment of the schema matching task by evaluating the baseline with newer models. Unfortunately, we couldn’t reproduce the data imputation task since the exact prompt was not accessible. In our experiment, the imputation task is selecting the missing labels from all existing labels in the training set. It is not clear how these classification labels were defined in the baseline implementation.

---

> ### Author Response · Authors · 2025-11-21
> **Comparison of agent frameworks using text to SQL use case**
>
> In the reubttal, we would like to show code examples to compare the programming models of Agentics, DSPy, and Langgraph.
> * DSPy is a very popular framework and pioneered the programming model framework for LLMs. DSPy uses Pydantic models (Signatures) to define data types and offers a PyTorch-like programming experience through Modules and Adaptors. We can write a program with a computational graph-like design for the workflows inside the Module.
> * LangGraph focuses on stateful agent orchestration using graph-based control flows.  It models workflows as nodes and edges, where each node represents an agent or tool and edges define message-passing dependencies.
> * Agentics offers a principled algebra for typed, composable workflows and fine-grained asynchronous execution. While DSPy supports asynchronous calls, its primary focus remains on declarative prompt optimization and compilation strategies.
> DSPy typically achieves asynchrony by compiling entire programs asynchronously or using an async alternative to forward.
> In contrast, Agentics enables functional-style asynchronous execution at the level of individual transformations through logical transduction and asynchronous map/reduce functions.
>
> ### Overall Comparison
> Instead of arguing that one framework is superior,  we aim to highlight the different design intents and goals that
> Agentics, DSPy, and LangGraph were created to achieve. Each framework reflects a distinct philosophy.
> * LangGraph and the LangChain ecosystem offer a mature codebase with many useful libraries for agent orchestration, tracing, and state management. In fact, Agentics leverages prompt functions provided by LangGraph.
> * DSPy pioneered declarative programming for LLM workflows and introduced the mechanism for constructing prompts from Pydantic types (Signatures), which remains a core innovation.
> * Agentics, by contrast, is designed for massive asynchronous LLM API calls that transduce strings formatted from one type to another.  It directly manages workloads through the AG meta-class,  which orchestrates asynchronous transductions at a fine-grained level.  This differs from DSPy, which organizes logic imperatively in Modules, and from LangGraph, which requires defining states and graphs for transformations.
>
> Due to this design, Agentics programs tend to be simpler and more concise,  with most effort focused on specifying types and composing pipelines, as shown in multiple examples in the paper.  Error handlings are also easier because the AG meta-class serves as a single control point for Pydantic transducers.
>
> In DSPy, execution is distributed across multiple Modules, and in LangGraph, across graph nodes.
> When a desired program can be implemented using DSPy’s predefined Modules, the overhead in programming is negligible.
> Similarly, LangGraph offers powerful features like state tracing, which is difficult in Agentics framework.
>
> In enterprise structured data workflows, the primary requirement is often the ability to process large batches of transformations quickly and efficiently under strict schema constraints.
>
> Next, we will present code examples comparing the composability and programming models of Agentics, DSPy, and LangGraph. The Python scripts were first written by hand and then added comments and unified the styles  using the Gemini-2.5 model.

---

> > ### Author Response · Authors · 2025-11-21
> > **Text to SQL in Agentics**
> >
> > All three implementations handle data types in a similar way using Pydantic models,  and we provide their asynchronous versions for fairness.
> > In the Agentics example, the AG class loads the BIRD-dev dataset and performs a single logical transduction from input fields to the output SQL query. Internally, the AG class maintains a list of typed states, and the PydanticTransducer generates transductions by applying prompt instructions to each state.
> > These transductions are executed asynchronously, enabling fine-grained parallelism and efficient error recovery.
> > For additional transductions or transformations,  we can extend the text-to-SQL pipeline as shown in Appendix C.1.
> > ```python
> > from pydantic import BaseModel, Field
> > from typing import Optional, List, Union
> > from agentics.core.agentics import Agentics as AG
> > import asyncio
> > import os
> > import json
> >
> > # Pydantic Data Model (State Definition)
> > class TextToSQLItem(BaseModel):
> >     """
> >     Defines the structure for a single Text-to-SQL example.
> >     This acts as the state for the Agentics self-transduction.
> >     """
> >     question: str = Field(..., description="Natural language question for SQL generation.")
> >     evidence: List[str] = Field([], description="Hints/Notes for writing the SQL query.")
> >     ddl_schema: str = Field(..., description="DDL script for database tables.")
> >     generated_sql_query: Optional[str] = Field(None, description="The generated SQL query.")
> >
> >
> > async def generate_sql_query(
> >     dataset: AG[TextToSQLItem],
> >     input_fields: List[str]
> > ) -> AG[TextToSQLItem]:
> >     """
> >     Uses Agentics self-transduction to translate the input fields (question, evidence, ddl_schema)
> >     into the desired output field (generated_sql_query).
> >     """
> >     output_fields = ["generated_sql_query"]
> >
> >     # Execute the self-transduction (the LLM call)
> >     # The instructions set on the dataset will be used here.
> >     dataset = await dataset.self_transduction(
> >         input_fields,
> >         output_fields,
> >         instructions=dataset.instructions # Pass instructions explicitly for clarity
> >     )
> >     return dataset
> >
> >
> > if __name__ == "__main__":
> >     # Load the data into an Agentics Dataset object
> >     text2sql_dataset = AG.from_json(
> >         "bird_dev_extended.jsonl",
> >         atype=TextToSQLItem,
> >         jsonl=True
> >     )
> >
> >     # Define Instructions
> >     # These instructions guide the LLM during the self-transduction step.
> >     text2sql_dataset.instructions = """
> >     You are an expert SQL writer. Your task is to translate the input natural language
> >     **question** to a valid SQLite query based on the provided **ddl_schema**.
> >     Use the **evidence** as hints. Your final output must *only* contain the SQL query.
> >     """
> >
> >     # Run the Transduction
> >     input_fields_to_use = ["question", "evidence", "ddl_schema"]
> >     final_dataset = asyncio.run(
> >         generate_sql_query(
> >             text2sql_dataset,
> >             input_fields_to_use
> >         )
> >     )
> > ```

---

> > > ### Author Response · Authors · 2025-11-21
> > > **Text to SQL in DSPy**
> > >
> > > A DSPy program is structured around three core components.
> > > Signatures, which define data types, Adapters, which encapsulate LLM API calls, and  Modules, which implement program logic. Similar to PyTorch, the forward method in a DSPy Module is where the workflow logic resides.
> > > This may include calling functions or chaining multiple LLM calls through other Modules.
> > > This design supports modularity and composability.  However, asynchronous execution in DSPy applies at the level of the forward method, meaning that fine-grained parallelism requires implementing each step as a separate Module.
> > >
> > > ```python
> > > import dspy
> > > import asyncio
> > > from typing import List
> > > from dspy.datasets import DataLoader
> > > from dspy import Signature, InputField, OutputField, Predict, Module, settings
> > > from dspy.adapters import ChatAdapter
> > >
> > > # Placeholder for the missing schema retrieval function
> > > def get_schema_description(db_id: str) -> str:
> > >     """Mock function to simulate getting the DB schema for a given ID."""
> > >     return f"""
> > >
> > > # Placeholder for the missing SQL cleanup function
> > > def cleanup_sql(sql_str: str) -> str:
> > >     """Simple cleanup to remove common LLM markdown wraps."""
> > >     sql_str = sql_str.strip()
> > >     if sql_str.lower().startswith("```sql"):
> > >         sql_str = sql_str[len("```sql"):].strip()
> > >     if sql_str.endswith("```"):
> > >         sql_str = sql_str[:-len("```")].strip()
> > >     return sql_str.replace('\n', ' ')
> > >
> > >
> > > # SIGNATURE
> > > class TextToSQLBasicSignature(Signature):
> > >     """You are an expert SQL writer and database analyst. Your task is to translate natural language QUESTION into a valid SQL query that can be executed against a given database schema."""
> > >     DB_SCHEMA: str = InputField(desc="DB table definitions showing names, columns, and relations.")
> > >     NOTE: str = InputField(desc="Hints for writing down the SQL query.")
> > >     QUESTION: str = InputField(desc="Natural language question to convert to SQL.")
> > >     SQL: str = OutputField(desc="Syntactically correct SQL with correct column names using suitable tables and rows. Your response should *only* contain the SQL query. Do not provide explanations or any other text.")
> > >
> > > # DSPy ADAPTER
> > > # This adapter is primarily used to post-process the LLM's output (cleanup).
> > > class SQLChatAdapter(ChatAdapter):
> > >     def parse(self, signature, completion):
> > >         # Use the base class parsing to get the output fields (e.g., {"SQL": "..."})
> > >         fields = super().parse(signature, completion)
> > >
> > >         # Apply the cleanup function to the SQL output field
> > >         if "SQL" in fields:
> > >             fields["SQL"] = cleanup_sql(fields["SQL"])
> > >         return fields
> > >
> > > # DSPy MODULE
> > > class TextToSQLPrompter(Module):
> > >     def __init__(self):
> > >         super().__init__()
> > >         self.program = Predict(TextToSQLBasicSignature)
> > >
> > >     async def aforward(self, db_id: str, question: str, evidence: List[str]):
> > >         """
> > >         The native asynchronous forward pass for the module.
> > >         Uses .acall() on the internal predictor.
> > >         """
> > >         table_descriptions = get_schema_description(db_id)
> > >         evidence_str = "\n".join(evidence)
> > >
> > >         # Use .acall() and await the result
> > >         prediction = await self.program.acall(
> > >             DB_SCHEMA=table_descriptions,
> > >             NOTE=evidence_str,
> > >             QUESTION=question
> > >         )
> > >         return prediction
> > >
> > > async def run_batch_aforward(model: TextToSQLPrompter, dev_set: List[dspy.Example]):
> > >     """Runs a batch of DSPy predictions concurrently using the module's aforward."""
> > >     print(f"Starting concurrent execution for {len(dev_set)} questions...")
> > >
> > >     tasks = []
> > >
> > >     for example in dev_set:
> > >         # Get the input dictionary for the module
> > >         inputs = example.inputs()
> > >
> > >         # Create an asynchronous task using the module's aforward method
> > >         async def aforward(model, inputs, original_q):
> > >             # Call the custom module's aforward
> > >             prediction = await model.aforward(**inputs)
> > >             return {
> > >                 "question": original_q,
> > >                 "generated_sql": prediction.SQL,
> > >                 "error": None
> > >             }
> > >         tasks.append(aforward(model, inputs, example.question))
> > >
> > >     # Run all tasks concurrently and wait for all to complete
> > >     results = await asyncio.gather(*tasks)
> > >     return results
> > >
> > >
> > > if __name__ == "__main__":
> > >     # Example setup using OpenAI
> > >     lm = dspy.OpenAI(model="llm model id",)
> > >     dspy.settings.configure(lm=lm, adapter=SQLChatAdapter())
> > >
> > >     # Load Data
> > >     data_loader = DataLoader()
> > >     dev_set = data_loader.from_json(file_path="bird_dev_extended.jsonl",
> > >                                           fields=["db_id", "question", "evidence"])
> > >     # Instantiate the Module and Evaluate
> > >     text2sql = TextToSQLPrompter()
> > >
> > >     # Run the full batch asynchronously using aforward
> > >     final_results = asyncio.run(run_batch_aforward(text2sql, dev_set))
> > > ```

---

> > > > ### Author Response · Authors · 2025-11-21
> > > > **Text to SQL in Langgraph**
> > > >
> > > > The LangGraph example demonstrates how an application is defined as a state graph,  starting with a Graph State and adding nodes for specific operations, such as generating SQL from a question using an LLM.
> > > > Compared to DSPy, LangGraph expresses program logic declaratively through graph nodes and edges, which is intuitive for multi-step workflows. Asynchronous execution is supported by adding asynchronous nodes.
> > > > However, parallelism is controlled at the graph level, not at the level of individual nodes.
> > > > In practice, the run-batch function loops over problems and invokes the application per state, meaning fine-grained parallelism is limited.
> > > > By contrast, Agentics enables pipeline-level control where each transformation (analogous to a node)
> > > > can execute asynchronously and independently, offering greater flexibility and scalability for structured workflows.
> > > >
> > > > ```python
> > > > import operator
> > > > import asyncio
> > > > from typing import TypedDict, Annotated, List, Dict
> > > >
> > > > # LangGraph and LangChain Imports
> > > > from langgraph.graph import StateGraph, END, START
> > > > from langchain_core.prompts import ChatPromptTemplate
> > > > from langchain_openai import ChatOpenAI
> > > >
> > > > # --- Graph State Definition ---
> > > > class TextToSQLState(TypedDict):
> > > >     """
> > > >     Represents the state of our graph.
> > > >     """
> > > >     db_schema: str
> > > >     question: str
> > > >     sql_query: Annotated[str, operator.replace]
> > > >     error: Annotated[str, operator.replace]
> > > >
> > > >
> > > > # --- Prompt Template ---
> > > > SQL_PROMPT_TEMPLATE = """You are an expert SQL writer and database analyst. Your task is to translate natural language QUESTION into a valid SQL query that can be executed against a given database schema.
> > > >
> > > > DB_SCHEMA:
> > > > {db_schema}
> > > >
> > > > QUESTION:
> > > > {question}
> > > >
> > > > NOTE:
> > > > You must only output the syntactically correct SQL query. Do not provide explanations or any other text.
> > > > """
> > > > sql_prompt = ChatPromptTemplate.from_template(SQL_PROMPT_TEMPLATE)
> > > >
> > > >
> > > > # --- Asynchronous Node for SQL Generation (The core change) ---
> > > > async def generate_sql_async(state: TextToSQLState) -> TextToSQLState:
> > > >     """
> > > >     Generates a SQL query from the question and schema using an LLM asynchronously.
> > > >
> > > >     NOTE: The function is now 'async def' and uses 'await llm.ainvoke'.
> > > >     """
> > > >     # Initialize your chosen LLM (LangChain clients are often thread-safe)
> > > >     llm = ChatOpenAI(model="gpt-4o", temperature=0)
> > > >
> > > >     # Format the prompt
> > > >     formatted_prompt = sql_prompt.format(
> > > >         db_schema=state["db_schema"],
> > > >         question=state["question"]
> > > >     )
> > > >
> > > >     # Run the LLM asynchronously
> > > >     try:
> > > >         response = await llm.ainvoke(formatted_prompt)
> > > >         sql_query = response.content.strip()
> > > >
> > > >         # Simple cleanup (removing markdown fences)
> > > >         if sql_query.lower().startswith("```sql"):
> > > >             sql_query = sql_query[len("```sql"):].strip()
> > > >         if sql_query.endswith("```"):
> > > >             sql_query = sql_query[:-len("```")].strip()
> > > >
> > > >         # print(f"Generated SQL for '{state['question'][:30]}...': {sql_query}") # Uncomment for debugging
> > > >
> > > >         # Update the state with the generated SQL
> > > >         return {"sql_query": sql_query, "error": ""}
> > > >
> > > >     except Exception as e:
> > > >         print(f"Error during SQL generation for: {state['question'][:30]}...: {e}")
> > > >         # Update state with the error
> > > >         return {"sql_query": "", "error": f"Generation failed: {e}"}
> > > >
> > > >
> > > > # --- Graph Construction ---
> > > > # Build the Graph
> > > > workflow = StateGraph(TextToSQLState)
> > > >
> > > > # Add the single Text-to-SQL node (using the async function)
> > > > workflow.add_node("sql_generator", generate_sql_async)
> > > >
> > > > # Define the single, linear path
> > > > workflow.add_edge(START, "sql_generator")
> > > > workflow.add_edge("sql_generator", END)
> > > >
> > > > # Compile the graph
> > > > app = workflow.compile()
> > > >
> > > >
> > > > # --- Asynchronous Batch Runner ---
> > > > async def run_batch(problems, app):
> > > >     """
> > > >     Runs the LangGraph app concurrently for a list of questions.
> > > >     """
> > > >     tasks = []
> > > >
> > > >     for problem in problems:
> > > >         # Prepare the initial state for the current question
> > > >         initial_state = {
> > > >             "db_schema": problem["db_schema"],
> > > >             "question": problem["question"],
> > > >             "sql_query": "",
> > > >             "error": "",
> > > >         }
> > > >         # Create an asynchronous task for each question using app.ainvoke()
> > > >         tasks.append(app.ainvoke(initial_state))
> > > >
> > > >     # Run all tasks concurrently and wait for them to complete
> > > >     concurrent_results = await asyncio.gather(*tasks)
> > > >
> > > >     # Process the results
> > > >     final_results = []
> > > >     for state in concurrent_results:
> > > >         final_results.append({
> > > >             "question": state['question'],
> > > >             "sql_query": state['sql_query'],
> > > >             "error": state['error']
> > > >         })
> > > >
> > > >     print("Concurrent execution finished.")
> > > >     return final_results
> > > >
> > > >
> > > > if __name__ == "__main__":
> > > >     # The asyncio.run() function is used to execute the top-level async function
> > > >     # assume that we loaded list of dictionaries storeing text-to-sql problems.
> > > >     final_results = asyncio.run(run_batch(problems, app))
> > > > ```

---

> ### Author Response · Authors · 2025-11-21
> **Additional Experiment Result from DiscoveryBench**
>
> The DiscoveryBench (https://github.com/allenai/discoverybench) dataset is a relatively recent benchmark designed to evaluate the full pipeline of data-driven scientific discovery. It contains 264 real-world discovery tasks spanning six scientific domains, including sociology, engineering, and various experimental sciences. Each domain is defined by structured data, accompanying metadata, and a natural-language discovery goal. To solve the DiscoveryBench tasks, it is crucial to interpret data semantically, identify salient variables, generate hypotheses, and compose multi-step reasoning workflows.
>
> Since the input of this dataset is in CSV files with numerical fields, many of the state of the art approaches rely on code generation and execution, namely the agentic AI systems produce Python code and execute it on the numeric heavy dataset, interpret the result, and refine the hypothesis. As similar to other structured data workflows, failure to constrain the type semantics results in brittle pipelines and limiting the scalability.
>
> In Agentic framework, we follow the schema (type) constrained composition of the data transformations with logical transduction and asynchronous MapReduce.
> Each row of raw dataset first maps to a typed object that extracts evidence via asynchronous Map transductions. Then, typed intermediate states are aggregated through asynchronous Reduce transduction that synthesize into a final state with Answer type.
>
> We evaluate the hypothesis match score (HMS) defined in the paper, an agent using GPTOSS-120B implemented in Agentics shows HMS score 21.4 in DB-REAL category, which is higher than other agents in the baseline using GPT-4o.
> The agent implemented in Agentics shows better performance than other agents such as CodeGen or React. The resulting pipeline processes all 285 discovery tasks under one hour with batch processing 20 parallel calls.
>
> | **Baselines** | **HMS** |
> | :--- | :---: |
> | NoDataGuess (GPT-4o) | 0.0 |
> | CodeGen (GPT-4o) | 15.5 |
> | React (GPT-4o) | 15.4 |
> | DataVoyager (GPT-4o) | 15.4 |
> | **Agentics (GPT-OSS120B)** | **21.4** |
> | Reflexion (Oracle) | 24.5 |

---

### Author Response · Authors · 2025-12-03
**Summary of Rebuttal**

Dear Area Chairs and Reviewers,

We sincerely thank the reviewers for their thoughtful and valuable feedback. This summary highlights how the concerns raised have been fully addressed in our rebuttal.


## Reviewer Consensus on Strengths
Reviewers recognized that Agentics represents a genuine paradigm shift in building LLM-based applications, or "agentic AI." The introduction of the formal Logical Transduction Algebra (LTA) is seen as a profound and novel contribution that ensures high-quality, rigorous work (jijG). This principled, well-executed approach reframes LLM agents as stateless transducers, effectively addressing the critical challenge of building robust, composable, and efficient LLM-based workflows (Bc8M/PX9e). Furthermore, the approach is strongly validated across multiple domains and tasks (Bc8M).


## Addressing Key Concerns
The main concerns focused on two areas: Conceptual motivation and Experimental comparison.

1. Conceptual Concerns (Motivation & Statelessness): Concerns were raised regarding the motivation for avoiding existing frameworks (Bc8M), the potential over-optimization for structured tasks (PX9e), and the limitations of the stateless paradigm (jijG).

* We provided a detailed comparison of Agentics, DSPy, and LangGraph via a concrete text-to-SQL pipeline. This demonstrated how Agentics leads to simpler, and declarative programs where composition and type specification are primary, and error handling is centralized via the $AG$ meta-class.
* To address the 'over-optimized' concern, we added a new experiment on DiscoveryBench, which requires complex semantic reasoning. Agentics using GPTOSS-120B outperforms existing baselines with GPT-4o models, demonstrating its strong potential for context-rich and open-ended tasks beyond traditional structured workflows.
* We addressed the limitation of stateful tasks by providing a general program pattern for designing transduction pipelines using functions and tools to manage and mutate external state.

2. Experimental Concerns (Framework Comparison & Baselines): Reviewers requested deeper comparisons with frameworks like DSPy and an update to older baselines (GPT-3.5) in schema matching and data imputation.
* Framework Comparison: We provided an in-depth code comparison in the global response, explaining the inherent difficulty of directly comparing programming models, while showcasing the benefits of Agentics in code structure.
* Updated Baselines: We replaced the GPT-3.5 baseline with GPTOSS-120B for schema matching and introduced the DiscoveryBench domain. These updates confirm that Agentics programs not only achieve better performance than the baselines but also demonstrate the key advantages of robustness, conciseness, and composability.


We believe we have addressed all concerns raised during the rebuttal phase, reinforcing the finding that Agentics offers a mathematically grounded, highly efficient, and empirically validated programming model for the rapidly evolving field of Agentic AI.
We commit to updating our draft with the new experimental results and the detailed explanations provided in this rebuttal (Bc8M/PX9e).
We respectfully request that the reviewers and Area Chairs consider updating the scores based on the additional discussions and new evidence during the remaining review period.

---

### Note · Program_Chairs · 2026-01-17
**Submission Desk Rejected by Program Chairs**

The following references in this submission do not refer to real documents and/or have major errors in bibliographic information:

 For Li et al., “MedQA: A dataset of medical question answering,” arXiv:2007.03233 (2020),

For Kwon et al., “Efficiently scaling transformer inference,” Proceedings of Machine Learning and Systems (2023)